# The genomic basis of the plant island syndrome in Darwin's giant daisies

José Cerca [1✉], Bent Petersen [2,3], José Miguel Lazaro-Guevara [4], Angel Rivera-Colón [5], Siri Birkeland[6,7], Joel Vizueta [8], Siyu Li[9], Qionghou Li[4], João Loureiro[10], Chatchai Kosawang[11], Patricia Jaramillo Díaz[12,13], Gonzalo Rivas-Torres[14,15,16,17], Mario Fernández-Mazuecos[18], Pablo Vargas[19], Ross A. McCauley[20], Gitte Petersen[21], Luisa Santos-Bay [2], Nathan Wales[22], Julian M. Catchen [5], Daniel Machado[23], Michael D. Nowak[7], Alexander Suh [24,25], Neelima R. Sinha [9], Lene R. Nielsen [11], Ole Seberg [26], M. Thomas P. Gilbert[1,2], James H. Leebens-Mack[27], Loren H. Rieseberg[4] & Michael D. Martin [1✉]

The repeated, rapid and often pronounced patterns of evolutionary divergence observed in insular plants, or the 'plant island syndrome', include changes in leaf phenotypes, growth, as well as the acquisition of a perennial lifestyle. Here, we sequence and describe the genome of the critically endangered, Galápagos-endemic species *Scalesia atractyloides* Arnot., obtaining a chromosome-resolved, 3.2-Gbp assembly containing 43,093 candidate gene models. Using a combination of fossil transposable elements, *k*-mer spectra analyses and orthologue assignment, we identify the two ancestral genomes, and date their divergence and the polyploidization event, concluding that the ancestor of all extant *Scalesia* species was an allotetraploid. There are a comparable number of genes and transposable elements across the two subgenomes, and while their synteny has been mostly conserved, we find multiple inversions that may have facilitated adaptation. We identify clear signatures of selection across genes associated with vascular development, growth, adaptation to salinity and flowering time, thus finding compelling evidence for a genomic basis of the island syndrome in one of Darwin's giant daisies.

A full list of author affiliations appears at the end of the paper.

As European naturalists set sail to explore the world, the distinctiveness of insular species stood out from the remaining biota. The collections carried out in the Galápagos, Cape Verde and Malay archipelagos were key for the development of the theory of natural selection[1] and biogeography[2]. More recently, Ernst Mayr's work, which set the scene for the modern synthesis[3], focused heavily on island biota[4]. The central role of remote archipelagos in our understanding of evolution is not coincidental. Organisms colonising these regions encounter highly distinct microenvironments that provide abundant ecological niches and thus ideal conditions for rapid and pronounced phenotypic change[5]. The 'island syndrome hypothesis' predicts the repeated and pronounced phenotypic shifts that some species undergo after colonising islands, as a result of a specific set of environmental conditions[6]. While island syndrome hypothesis has been well established[6,7], its integration with genomic evidence still lags. For instance, the changes in body size observed in insular animal lineages, when compared to their continental counterparts, are the textbook example of an island syndrome (e.g., pygmy mammoths and giant tortoises), however, to the extent to which these changes are hereditary (genetic) or induced by different food sources (diet) has yet to be documented for many lineages. Considering the rapid nature of these changes, it can be expected that rearrangements in genome structure contribute to the adaptation to novel environmental conditions.

Because the most prominent examples of island syndromes feature animal lineages, our understanding of these phenomena in plants lags[7]. As plants colonise archipelagos, they typically undergo shifts in leaf phenotypes, overall size, woodiness, lifespan and have an altered dispersal ability—the plant island syndrome[7]. This is well exemplified by the iconic, yet understudied, daisies in the genus *Scalesia*[8–11]. This group consists of ca. 15 species, which have colonised moist forests, littoral zones, arid zones, dry forests, volcanic soils, lava gravels and fissured environments across varied elevations[8,12]. The phenotypic changes undergone by *Scalesia* include an increased woodiness, leaf-morphology variation, simplified inflorescences, increased growth rates, and gigantism—as expected by the plant island syndrome. Indeed, this outstanding phenotypic and ecological variation has led authors to refer to this group as the 'Darwin finches of the plant world'[13]. All *Scalesia* species are ancestrally tetraploid ($2n = 4x = 68$)[14,15], and the polyploid genetics may have provided the genetic grist for the diversification, as suggested for island floras[16].

In this work, we describe a high-quality chromosomal reference genome assembly and annotation for *Scalesia atractyloides*, an endemic plant to the Santiago Island, Galápagos. This species was selected because it is critically endangered and has low genomic heterozygosity suitable for de novo assembly[9]. A chromosome-resolved assembly has allowed us to identify and separate the two ancestral genomes that united in the polyploidization event, and to compare gene and transposable element distribution across and between these subgenomes. Annotation of genes using PacBio IsoSeq RNA afforded a high-quality annotation of the genome, and the detection of selection and gene-family expansions that implicate the genomic basis for island syndrome traits in this charismatic group of plants.

## Results and discussion
**Genome assembly, annotation and quality control.** The *Scalesia atractyloides* genome assembly is highly contiguous (Fig. 1A), consisting of 3,216,878,694 base pairs (3.22 Gbp) distributed over 34 chromosome models, in line with previous cytological evidence[11,17,18]. The $N_{90}$ was of 31, corresponding to all but the three smallest chromosomes ($n = 34$) and $L_{90}$ was 81.66 Mbp.

Flow cytometry estimates (Supplementary Information; Supplementary Table 01), however, suggest a genome size of ca. 3.9 Gbp, and thus ~700 Mbp were likely collapsed by the assembler or removed by *purgehaplotigs*[19]. Despite the likely collapse of repeats, we were able to annotate 76.22% of the genome as repeats, which were masked by *RepeatMasker* (~2.5 Gbp). Considering the whole genome, 47.9% of the genome was composed of long-terminal repeat (LTR) retroelements, of which 16.2% were Copia and 31.54% Gypsy elements (Supplementary Information; Supplementary Table 02), and 26.32% were unclassified repeats.

The IsoSeq transcriptome recovered 46,375 genes and 224,234 isoforms (Supplementary Information; Supplementary Fig. 01). Using this RNA as evidence and ab initio models, we retrieved 43,093 genes from the annotation. Of the 430 Viridiplantae odb10 BUSCO groups used in a search of the genome (Fig. 1B), 401 were found as complete (93.3%), of which 245 were found as duplicate (57%) and 156 as a complete and single copy (36.3%), and 12 as fragmented (2.8%). Only 17 were absent (3.9%). When running OrthoFinder including *Scalesia* and five other Asteraceae chromosome-resolved assemblies, we found that 34% of all the orthogroups included genes from the five genomes, indicating a high-quality gene annotation (Supplementary Information; Supplementary Fig. 02). The proportion of annotated repeats and the number of genes is within the variation reported for Asteraceae. As an example, the closely related sunflower (*Helianthus annuus*) reference genome includes ~52,000 protein-coding genes and has a repeat content of 74%[20] the lettuce genome includes 74% repeats and ~39,000 protein-coding genes[21] and the Hawaiian *Bidens* genome has 70–74% repeats[22].

**Subgenome identification and evolution.** The identification of subgenomes (subgenome A and subgenome B; Fig. 2A) was carried out in two steps. In the first step, we assigned the 34 chromosomes into 17 homeolog pairs by identifying and mapping 1061 duplicated conserved orthologous sequences (COS; Supplementary Information; Supplementary Table 03). While this first step identified chromosome pairs (homeologs), it did not facilitate the assignment of subgenome identity within pairs. Homeolog exchanges[23] are therefore not a concern at this point. In the second step, we used the $k$-mer spectrum to identify 'fossil transposable elements' that were actively replicating while the two subgenomes were separated (i.e. before the polyploidization event). Since different genomes accumulate different transposable elements, we hypothesised that some transposable elements should be differentially distributed in different subgenomes[24,25]. In short, transposable element families active before the divergence of the two ancestral lineages (lineages A and B) are predicted to be approximately equally represented in both subgenomes, whereas transposable elements active after the divergence of the parental species and before the polyploidization event, are predicted to be differently represented within subgenomes. Using the $k$-mer spectrum, we selected 13-mers that: (i) were highly abundant, specifically, present at least 100 times in the genome. By selecting highly abundant genomic regions, we obtain genomic regions representing repeats/TEs; (ii) were unevenly represented between chromosome pairs (identified in the previous step). By selecting differentially represented 13-mers, we obtained a set of TEs/repeats which were active during the separation period (ancestral lineages A and B). were obtained using Jellyfish and a combination of in-house scripts, and we ended up with an average of 361 13-mers per chromosome pair (max = 934, min = 182, SD = 179). Using this selection of 13-mers, we ran a hierarchical clustering algorithm that grouped

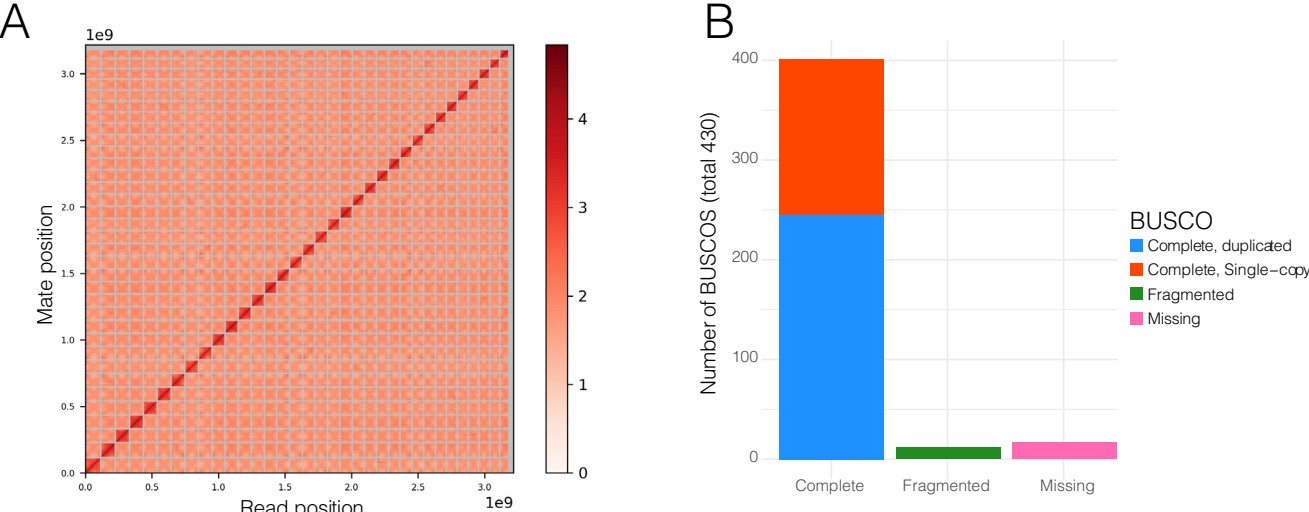

**Fig. 1 Chromosome-resolved assembly of the *Scalesia atractyloides* nuclear genome. A** Link density histogram, with 34 linkage groups (chromosome models) identified by contiguity ligation sequencing. The *x* and *y* axes show mapping positions of the first and second read in read pairs. **B** Viridiplantae BUSCO set, which offers a characterisation of conserved orthologue genes.

chromosomes into two clusters (two subgenomes; Supplementary Information; Supplementary Fig. 04). To clarify these assignments and, in particular, the occurrence of differentially represented transposable element families, we explored the output from *RepeatMasker* by plotting transposable element families unevenly represented across subgenomes (Fig. 2B, Supplementary Data 4 and Supplementary Fig. 05). The identification of differently represented transposable element families further provides compelling evidence that the *Scalesia* radiation is of allopolyploid origin, confirming chromosome counts[15]. Island floras are characterised by a high frequency of paleoallopolyploids (i.e. old allopolyploids)[26], and the genetic variation made available at higher ploidy levels may underpin the diversification to multiple environments (Julca et al.[27]; te Beest et al.[28])—a scenario which is in line with the evolutionary history of *Scalesia*.

Using four other chromosome-level assemblies from Asteraceae (*Helianthus annuus*, *Conyza canadensis*, *Mikania micrantha* and *Lactuca sativa*) and the two *S. atractyloides* subgenomes, we estimated groups of orthologous genes using OrthoFinder. We obtained 710 orthogroups in which each genome had only a single member, tolerating no missing data, and used this data to construct a phylogenetic tree. The tree topology agrees with the placement of the Asteraceae lineages from a recent and comprehensive set of genomic analyses[29]. We performed two separate dating analyses: one to date the nodes of the tree (speciation events) and another to date the polyploidization event. To date the nodes of the tree, we constrained the node separating the *Scalesia* subgenomes and *Helianthus* at 6.14 Mya (Fig. 2C) following recent literature[289]. Consistent with comparisons of Ks distributions (Supplementary Information; Supplementary Fig. 06), a model-based divergence time estimate (r8s) suggests that the ancestral lineages represented in the *Scalesia* subgenomes diverged at approximately 4.14 Mya. A second dating analysis was performed to date the polyploidization event using LTR retrotransposons. For this analysis, we used LTR retrotransposons which were evenly represented between subgenomes (to capture families active after the polyploidization event), and which were present in *Helianthus* (Supplementary Information; Supplementary Fig. 07). By comparing genetic divergence (Jukes Cantor distances) between the *Scalesia* LTR retrotransposons and *Helianthus*, we estimated that the ancestral genomes reunited in a single-polyploid genome at least 3.76 Mya (Fig. 2C; Supplementary Information; Supplementary Fig. 06).

These dates are concordant with the PSMC analysis which roughly indicate that the three *Scalesia* species had concordant population sizes of 250,000–300,000 circa 4 Mya (Fig. 2D). Mismatches between the three genomes could result from variation in generation time in *Scalesia*, and bottlenecks suffered by populations as a result of climatic shifts in the Galápagos[30]. The PSMC estimates are concordant with a recent dating analysis that estimated the divergence between *Pappobolus* and *Scalesia* occurred ~3 Mya[9].

The identification of subgenomes allowed comparing gene and transposable element distribution across homeolog chromosome pairs. We find that gene density is highest near the telomeres on both subgenomes, while transposable elements are more evenly distributed throughout chromosomes (Fig. 3A). This even distribution of transposable elements is different from most other vascular plants, in which transposable element load is highest near the centre and decreases towards the ends of the chromosome, and rather is reminiscent of observations in bryophyte genomes[31–33]. Even distributions of transposable elements were also observed in the sunflower genome[20], which may be indicative of particular transposable element regulation in the Heliantheae. However, these patterns should be confirmed as more Heliantheae genomes are sequenced.

As two genomes unite to form a single hybrid genome, an accommodation of the two subgenomes, the process of diploidization takes place[34–36]. This process can occur very quickly, with changes in transcription between subgenomes observed in 2–3 generations[37], and result in pronounced changes in gene numbers. Whereas subgenome dominance in gene expression and retention has been documented in paleopolyploid plant genomes[38–40], *Scalesia* subgenomes contain roughly equal gene and isoform contents (Fig. 3B, C), as well as pseudogene numbers and transposable element load (Fig. 3D, E). In addition to this, when running the Viridiplantae BUSCO set for each subgenome separately, we find 82.7% complete BUSCOs on subgenome A (76.6% single copy, 6% duplicates), and 81.9% complete BUSCOs (77% single copy, 4.9% duplicates) on subgenome B. Both subgenomes are roughly the same length (subgenome A = 1,629,251,263 bp; subgenome B = 1,554,170,668 bp), and have retained the same number of chromosomes (Fig. 3A). This indicates that during the past ~3.76 million years, during which the two subgenomes have been unified in the same organism, there has not been a drastic rearrangement of

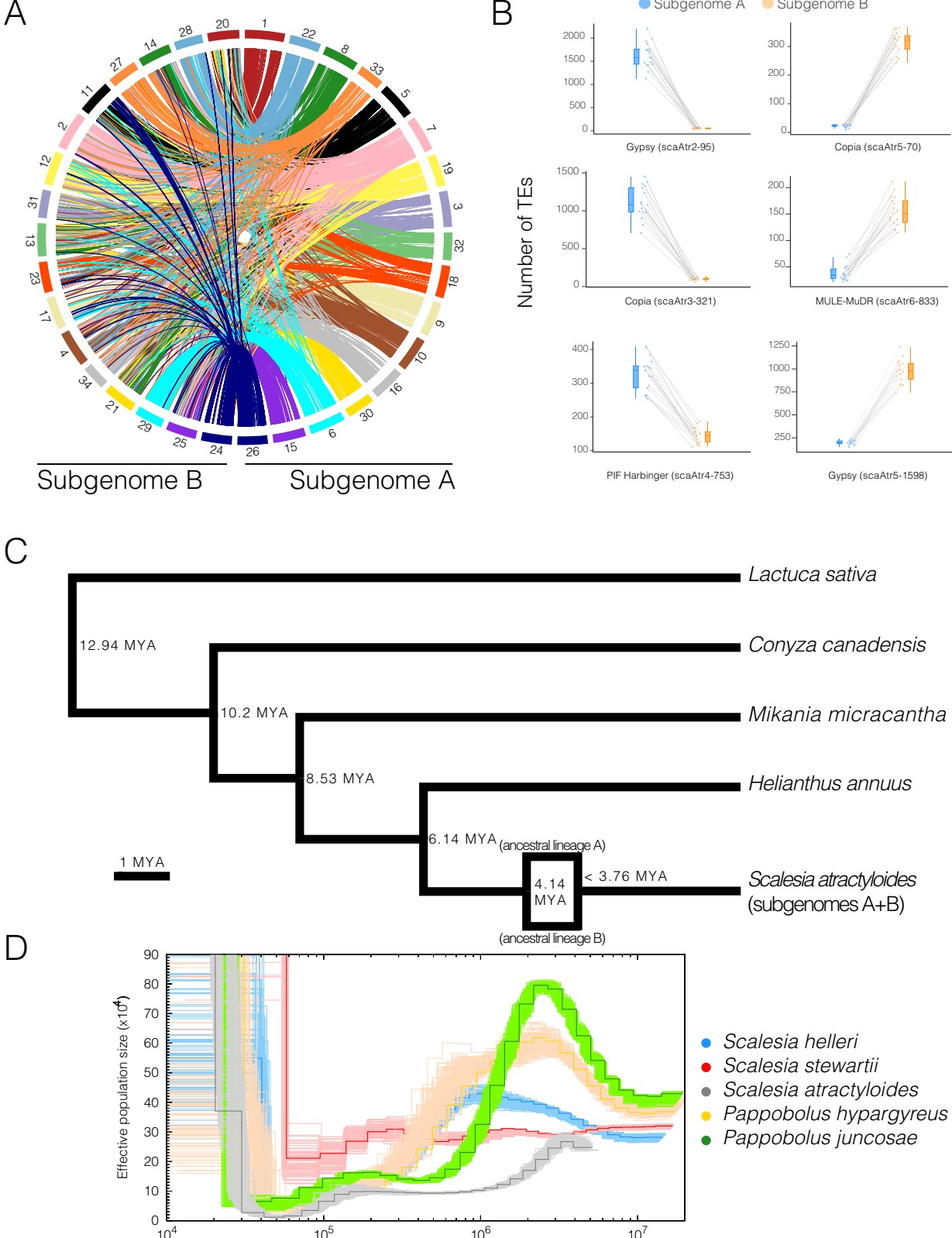

either subgenome relative to the other, despite a smaller accumulation of genes and pseudogenes on subgenome A. This suggests diploidization is slowed down and, to explain this, we speculate that *Scalesia*'s adaptation to insular environments has benefited from the genetic variation and diversity stemming from the allopolyploidization event[41].

**Genome rearrangements in Heliantheae.** To further dissect the mode and tempo of polyploid subgenome evolution, we used *Synolog*[42] to create chromosome stability plots, which allow us to detect translocations and inversions (Fig. 4). *Synolog* establishes clusters of conserved synteny by identifying single-copy orthologs shared between two genomes via reciprocal all-by-all BLAST.

**Fig. 2 Subgenomes and evolutionary history of *Scalesia*. A** Circos plots displaying the 34 chromosome models in the assembly. Pairs are organised to the left and right from the top, and have the same colour coding. **B** Selected families of transposable elements (TEs) that are differently represented on each subgenome (17 pairwise comparisons; each TE is labelled after the assembly (scaAtr), round and family number (2–95) after RepeatMasker). These TE families were likely active while the two subgenomes were separated, being thus unevenly represented. These highlight subgenome identification. Each data point corresponds to a chromosome in a subgenome (subgenome A in blue and B in orange). Chromosome pairs are linked by grey lines. **C** Single-copy ortholog phylogeny of the studied Asteraceae genome assemblies. Node ages are provided to the right of each node, as well as the predicted time for the polyploidization event. **D** Pairwise sequentially Markovian coalescent (PSMC) estimation of the demographic history of *Scalesia atractyloides*, two other *Scalesia* species, and two members of the *Pappobolus* genus, which is the sister taxon to *Scalesia* using bootstrapping (100 replicates). PSMC runs for the whole genome and subgenomes yielded similar results (Supplementary Fig. 03).

From the identified synteny clusters, we calculated statistics on the orientation (Forward/Inverted) and chromosome location. We thereby classified genes into four categories: 'Forward pair' (FP; i.e. not inverted, and the single-copy orthologs are in chromosomes from the same pair), 'Inverted pair' (IP; i.e. inverted, and the single-copy orthologs are in chromosomes from the same pair), 'Forward translocated' (FT; i.e. not inverted, and the orthologues are not in chromosomes from the same pair), 'Inverted translocated' (IT; i.e. inverted, and the orthologues are not in chromosomes from the same pair). Comparing the two *Scalesia* subgenomes, we found 4379 FP genes (comprising 111 clusters of 5 or more genes), 5642 IP genes (78 clusters), 747 FT genes (31 clusters), and 1488 IT genes (18 clusters), totalling 12,256 genes included in the analysis (Fig. 4B). In terms of genome length, we classified 1.45 Gbp as FP, 1.15 Gbp as IP, 346.4 Mbp as FT and 343.3 Mbp as IT. Thus, while the majority of the genes have been inverted (7130 genes), a minor fraction of the genome length has been inverted (1.49 Gb). Despite the fact that we were able to identify homeologs and the subgenomes, the synteny plots confirm there are rapid rates of chromosomal rearrangements in the Asteraceae[20], and suggest a central role of inversions in the family.

**Evidence for the island syndrome**. We identified 920 genes under selection ($P < 0.05$) in the *Scalesia* genome (478 on subgenome A and 442 on subgenome B), after correcting dN/dS ratios using a Holm–Bonferroni FDR correction. To understand their function we took two approaches, one generalistic using GO enrichment analysis, and a more detailed one where we randomly selected 100 genes (Supplementary Data 1), called *Arabidopsis* orthologs and read the literature for those genes. Before this analysis, we confirmed that the selection of 100 random genes did not bias the final results by comparing a GO analysis using the 920 genes (Fig. 5A) with a GO analysis with only 100 genes. Subsampling did not bias the major categories. First, we extracted the functional annotation using a Gene Ontology (GO) term enrichment analysis, and the results were visualised using *Revigo*. *Revigo* organised GOs onto groups, which we coloured and named: metabolic processes (Fig. 5A, orange group), cellular reorganisation (green group), DNA repair (yellow group), response to protein folding (maroon group), and regulation (regulation of metabolic processes, translation, gene expression, translation, nuclear division, chromosome segregation, among others; pink group; Fig. 5A; Supplementary Information; Supplementary Data 2 and 3). Genes inferred to have evolved under positive selection are also associated with meiosis, chromosome arrangement and chromatin status (meiotic cytokinesis, the establishment of chromosome location, chromosome separation and chromosome segregation, among other GO classifications; Fig. 5A, Supplementary Information; Supplementary Data 2 and 3), and this may indicate selection at genes associated with the coexistence of two genomes.

Considering the astonishing leaf phenotypic variation in the *Scalesia* lineage, it is particularly interesting that we detected selection on potential regulators of leaf morphology, including genes well known to determine leaf cell number in *A. thaliana*

(E2F1)[43,44], cell fate in leaves (YABBY5)[45,46], leaf senescence (RANBPM, LARP1C, PEN3)[47–49], leaf variegation (THF1)[50,51] and leaf growth (PAC)[52–54]. It has been observed that *Scalesia* plants grown in shaded conditions, as opposed to the continuous direct light provided by their open Galápagos landscape habitats, show substantially retarded growth[55]. Thus it is interesting to note that many *Scalesia* genes under selection are affected by light stimulus. A STRING analysis showed that there was a selection at multiple points in the light regulatory pathways including responses to R/FR and blue light responses (Supplementary Fig. 08). These include an inhibitor of red and far-red light photoreceptor (PHL)[56,57], a lysine-tRNA ligase that regulates photomorphogenic responses[58], an amino acid aminotransferase-like PLP-dependent enzymes superfamily protein that is regulated under light conditions and is associated with the photorespiration process[59,60], and genes for which knock-out mutants experience alterations in light reception (DJC69, COX15; Supplementary Data 1)[61,62].

Many of the stress-response genes under selection in *Scalesia* are associated with osmotic stress in *A. thaliana*, concomitant with evidence that the *Scalesia atractyloides* habitat is characterised by arid conditions such as the Galápagos' arid zone, littoral zone, and fissured lava areas[8,12]. For instance, we identified selected genes ('Leucine-rich repeat protein kinase family protein', MPPBETA, 'leaf osmotic stress elongation factor 1-β-1', AT2G21250, VAP27-1) associated with osmotic stress[63–68], as well as heat shock proteins and regulators of stomatal closure (THF1)[50,51] (Supplementary Data 1). Other stress-associated genes under selection include those involved in response to high irradiation (ZAT10, AT1G06690, DDB2)[69–74].

Some genes under selection are associated with growth and transitions between life stages. *Scalesia* plants' fast rates of growth have earned them the name 'weedy trees', and these genes may regulate these plants' exceptionally fast growth and tree-like habits. We find three genes under selection that cause the transition between embryonic and vegetative traits (RING1A, SWC4, ABCI20)[75], and four genes that regulate flowering time in *A. thaliana* (ELF8, RING1A, Short-Vegetative-Phase, NRP1)[76–80], and height or size of the plant (CLAVATA, GH9C2, ELF8, NSL1, TUA6)[81–90].

Finally, we assessed the expansion and contraction of gene families in the *Scalesia* genome, finding a total of 37 significantly contracted families and 26 significantly expanded families (Fig. 5B). GO enrichment testing of the expanded families uncovered significantly enriched functions associated with vascularisation (secondary cell wall biogenesis, shoot system development, negative regulation of organ growth, xylem vessel member cell differentiation, protoxylem development), likely associated with plant growth in *Scalesia*[9]. We also find evidence of evolutionary responses to aridity and changes in osmotic pressure in significantly expanded families (regulation of stomatal closure, response to water deprivation, response to osmotic stress, water homoeostasis), similar to the genes under selection (Fig. 5B). Interestingly, we detect contraction in gene families with GO terms associated with tree

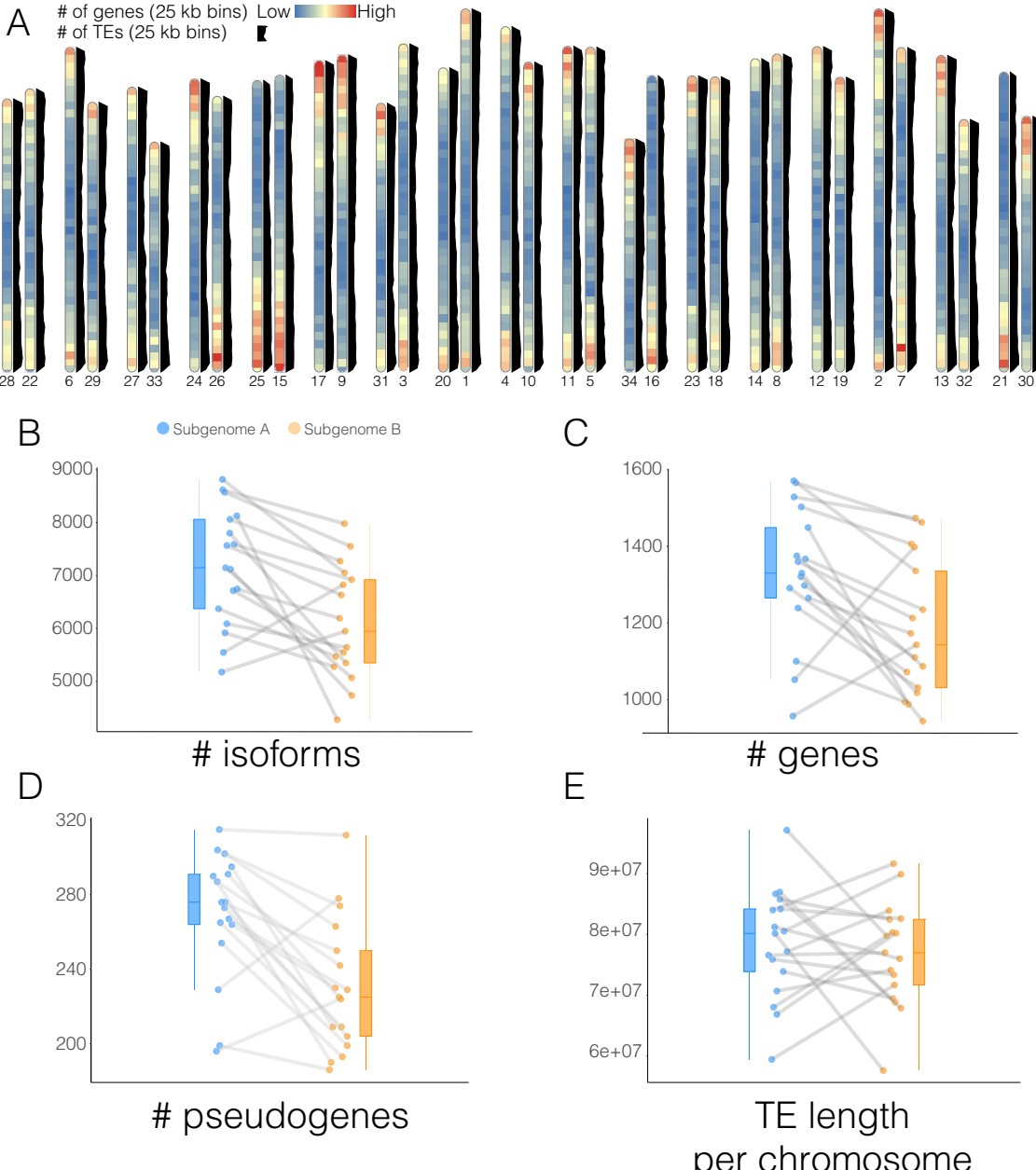

**Fig. 3 Subgenome evolution and characterisation. A** Ideogram with gene and transposable element distribution in *25-kbp* bins. Gene density is plotted in chromosome representations and transposable element distribution is plotted to the side of each chromosome in black. Chromosomes are arranged in homoeologous pairs. **B** The number of isoforms detected for each subgenome. Each data point corresponds to a chromosome in a subgenome (subgenome A in blue and B in orange; 17 pairwise comparisons; subgenome A: min = 5176; max = 8817, avg = 7175; Subgenome B: min = 4277; max = 7982, avg = 6103). Chromosome pairs are linked by grey lines. **C** Number of genes detected for each subgenome (17 pairwise comparisons; subgenome A: min = 1571; max = 1571, avg = 1327; Subgenome B: min = 944; max = 1474, avg = 1181). **D** The number of pseudogenes detected for each subgenome (17 pairwise comparisons; Subgenome A: min = 196; max = 315, avg = 270; subgenome B: min = 186; max = 312, avg = 230). **E** Length of transposable elements detected for each subgenome (17 pairwise comparisons; subgenome A: min = 59,440,194; max = 97,142,619, avg = 78,545,717; Subgenome B: min = 57,631,669; max = 91,646,014, avg = 76,870,839).

habits (shoot system development, regulation of organ growth, regulation of root development, xylem vessel member cell differentiation, gravitropism), adaptation to arid environments (water deprivation, stomatal closure, regulation to osmotic stress) and cold tolerance (cellular response to cold; Supplementary Information; Supplementary Tables 09–14). While this may seem contradictory, it suggests that different families have redundant functions, and the expansion of a family may lead to redundancy in another family and consequent gene loss through the pseudogene formation.

In this study, we were able to elucidate patterns of genome evolution in a critically endangered species (*Scalesia atractyloides*) of Darwin's giant daisy tree radiation by attaining a chromosome-resolved genome and by subsequently identifying two ancient genomes underlying its polyploid state. We found that both subgenomes retain a relatively similar number of genes as well as other genetic features, such as pseudogenes and transposable elements, which lead us to speculate on the role of insular evolution underlying these changes. Moreover, we uncovered the role of inversions in gene accumulation, suggesting these may

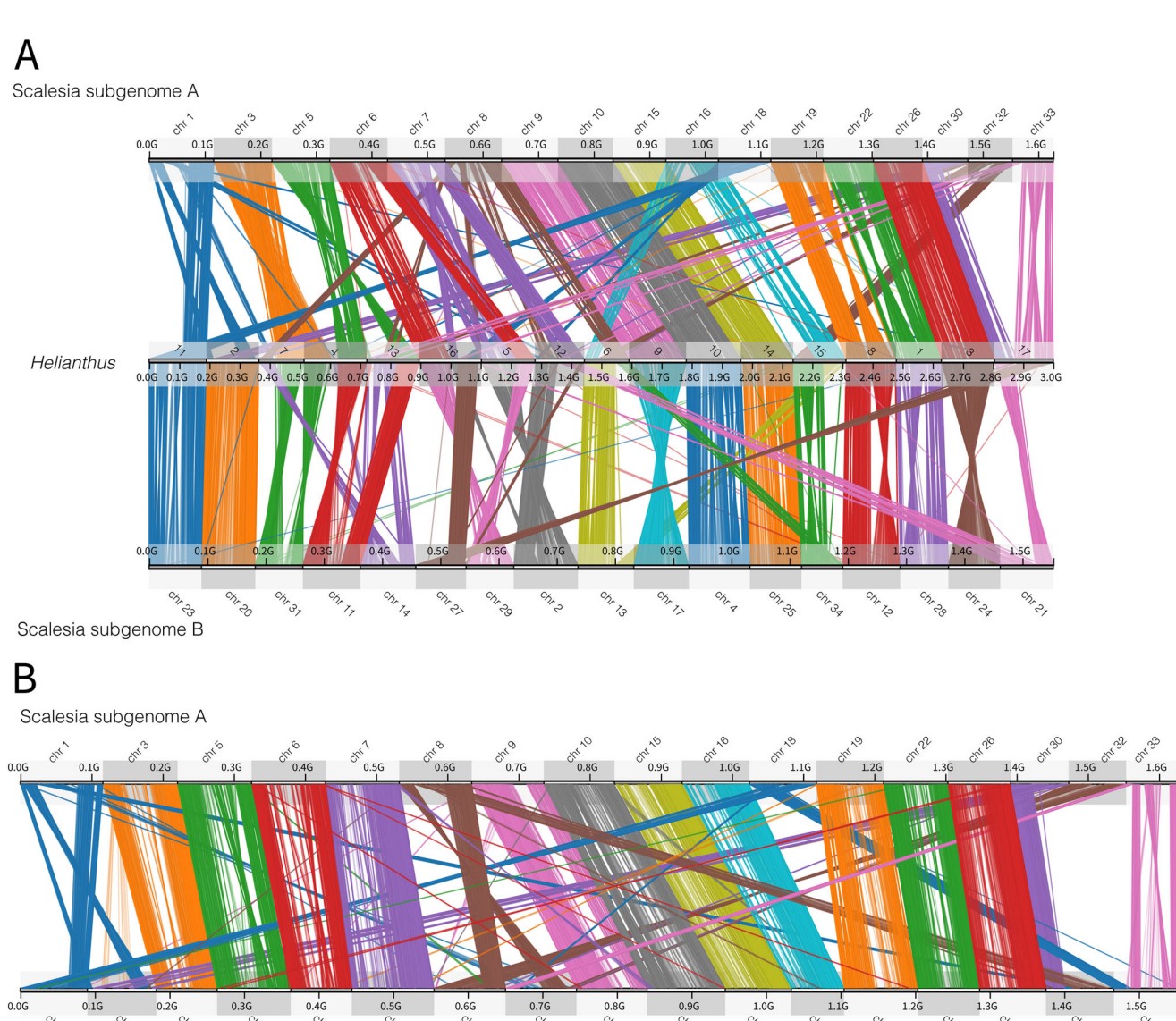

**Fig. 4 Chromosome stability plots.** These reveal the role of inversions and translocations in the differential in each subgenome. **A** Chromosome stability plot between the two *Scalesia* subgenomes and the *Helianthus annuus* genome. Each line connects a pair of orthologous genes, colour-coded by chromosome pair. **B** Chromosome stability between the two *Scalesia* subgenomes. Each line connects orthologous genes in subgenomes A and B, colour-coded by chromosome pair.

have played an important role in the maintenance of genes in subgenomes, and found a relatively unique pattern of transposable element accumulation within flowering plants which warrants further attention. Expanded gene families and genes under positive selection indicate the first solid evidence for genomic island syndrome in a plant, revealing an underlying genomic basis of the outstanding leaf and growth phenotypic variation in *Scalesia*. This phenotypic variation may also have been facilitated by the substantial presence of transposable elements and by ploidy.

## Methods

**Plant material, flow cytometry, DNA extraction, library preparation and sequencing.** Tissues used for the de-novo genome assembly and annotation were sampled from living *Scalesia atractyloides* plant P2000-5406/C2834 cultivated in the greenhouse of the University of Copenhagen Botanical Garden collections. This plant was originally germinated from a seed collected from Santiago Island. Fresh tissue was collected and flash-frozen in dry ice or liquid nitrogen and then stored at −80 °C for later use.

To assist with sequencing coverage strategy and to inform genome assembly, we obtained estimates of genome size using flow cytometry following[91]. Briefly, 50 mg of freshly collected leaves from the sample material and from the reference standard (*Solanum lycopersicum* 'Stupické'; 2 C = 1.96 pg[92]; were chopped with a razor blade in a Petri dish containing 1 ml of Woody Plant Buffer[93]. The nuclear suspension was filtered through a 30-μm nylon filter, and nuclei were stained with 50 mg ml$^{-1}$ propidium iodide (PI) (Fluka, Buchs, Switzerland). Fifty mg ml$^{-1}$ of RNase (Sigma, St Louis, MO, USA) was added to the nuclear suspension to prevent staining of double-stranded RNA. After a 5-min incubation period, samples were analysed in a Sysmex CyFlow Space flow cytometer (532 nm green solid-state laser, operating at 30 mW). At least 1300 particles in G1 peaks were acquired using the FloMax software v2.4d[94]. The average coefficient of variation for the G1 peak was below 5% (mean CV value = 2.72%). The holoploid genome size in mass units (2 C in pg; *sensu*[95] was obtained as follows: sample 2 C nuclear DNA content (pg) = (sample G1 peak mean/reference standard G1 peak mean) * genome size of the reference standard. Conversion into basepair numbers was performed using the factor: 1 pg = 0.978 Gbp[96]. Three replicates were performed on two different days, to account for instrumental artefacts.

The commercial provider Dovetail Genomics extracted and purified high-molecular-weight DNA from flash-frozen leaf tissue using the CTAB protocol, and the concentration of DNA was measured by Qubit. For long-read sequencing, they constructed a PacBio SMRTbell library (~20 kb) using the SMRTbell Template Prep Kit 1.0 (PacBio, CA, USA) following the manufacturer recommended protocol. This

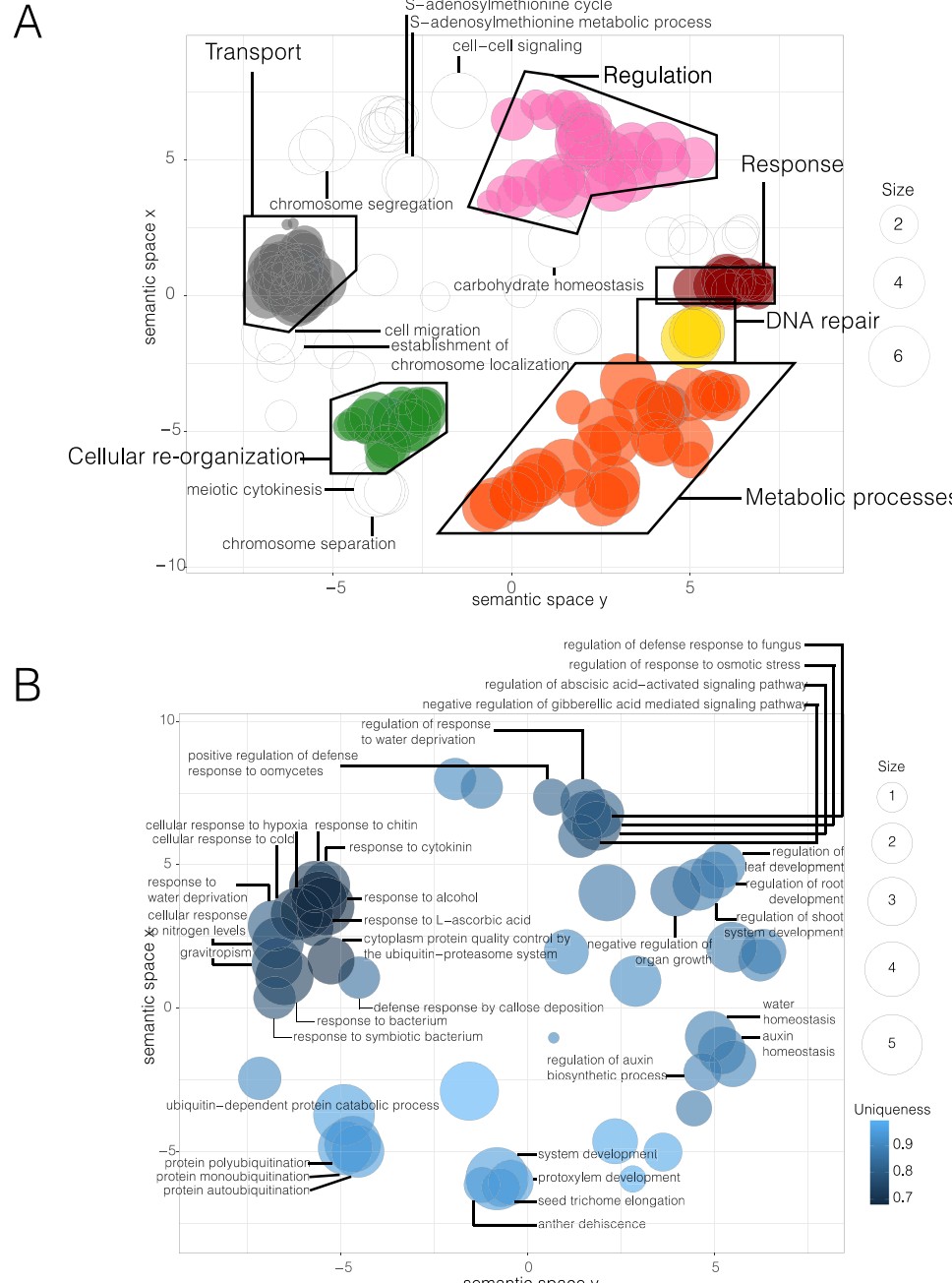

**Fig. 5 Positive selection and gene-family expansion across the *Scalesia atractyloides* genome. A** GO term enrichment of the genes under selection across the genome. GO terms assigned to at least four genes are labelled. Size refers to the number of genes associated with a particular GO term. **B** GO term enrichment of the genes belonging to expanded gene families across the genome according to a *CAFE* analysis. Only GOs within a group of three or more overlapping circles are included. Uniqueness measures the degree to which a particular GO term is distinct relative to the whole list.

library was bound to polymerase using the Sequel Binding Kit 2.0 (PacBio) and loaded onto the PacBio Sequel sequencing machine using the MagBeadKit v2 (PacBio). Sequencing was performed on the PacBio Sequel SMRT cell, using Instrument Control Software v5.0.0.6235, Primary analysis software v5.0.0.6236, and SMRT Link Version 5.0.0.6792. PacBio sequencing yielded 41,322,824 reads, resulting in a total of 197-fold coverage of the nuclear genome. For contiguity ligation, they prepared two Chicago libraries as described in ref. [97]. Briefly, for each Dovetail Omni-C library, chromatin is fixed in place with formaldehyde in the nucleus and then extracted. Fixed chromatin was digested with DNAse I, and chromatin ends were repaired and ligated to a biotinylated bridge adapter followed by proximity ligation of adapter-containing ends. After proximity ligation, crosslinks were reversed and the DNA was purified. Purified DNA was then treated to remove biotin that was not internal to ligated fragments. Sequencing libraries were generated using NEBNext Ultra enzymes and Illumina-compatible adapters. Biotin-containing fragments were isolated using streptavidin beads before PCR enrichment of each library. These

libraries were then sequenced on an Illumina HiSeq 2500 instrument, producing a total of 1,463,389,090 sequencing reads.

To obtain RNA transcript sequences for annotation of the genome, we extracted RNA from five tissues (root, stem, young leaf, old leaf, and floral head) of a *S. atractyloides* plant P2000-5406/C2834 using a Spectrum Plant Total RNA Kit (Sigma, USA) with on-column DNA digestion following the manufacturer's protocol. RNA extracts from all five tissues were pooled. mRNA was enriched using oligo (dT) beads, and the first-strand cDNA was synthesised using the Clontech SMARTer PCR cDNA Synthesis Kit, followed by first-strand synthesis with SMARTScribeTM Reverse Transcriptase. After cDNA amplification, a portion of the product was used directly as a non-size-selected SMRTbell library. In parallel, the rest of the amplification was first selected using either BluePippin or SageELF, and then used to construct a size-selected SMRTbell library after size fractionation. DNA damage and ends were then repaired, followed by hairpin adaptor ligation. Finally, sequencing primers and polymerase were annealed to SMRTbell templates,

and IsoSeq isoform sequencing was performed by Novogene Europe (Cambridge, UK) using a PacBio Sequel II instrument, yielding 223,051,882 HiFi reads.

**Genome assembly and annotation.** An overview of the bioinformatic methods is provided in https://github.com/jcerca/Papers/tree/main/scalesia_genome. The genome was assembled using *wtdbg2*[98], specifying a genome size of 3.7 Gbp, PacBio Sequel reads, and minimum read length of 5,000. The *wtdbg2* assembly consisted of contigs with 3.62 Gbp total length. This assembly was then assessed for contamination using *Blobtools* v1.1.1[99] against the NT database, detecting and removing a fraction of the scaffolds. This filtered assembly was used as input to *purge_dups* v1.1.2, which removed duplicates based on sequence similarity and read depth[100], reducing the assembly length to 3.22 Gbp. This assembly and the Dovetail Omni-C library reads were used as input data for *HiRise* by aligning the Chicago library sequences to the input assembly. After aligning the reads to the reference genome using *bwa*, *HiRise* produces a likelihood model for genomic distance between read pairs, and the model was used to identify misjoints, prospective joints, and make joins. After *HiRise* scaffolding, the $N_{50}$ increased to 16, and the $N_{90}$ to 31, corresponding to all but the three smallest chromosomes ($n = 34$), while the $L_{50}$ was 94.2 Mbp and the $L_{90}$ was 81.66 Mbp. The largest scaffold was 116.23 Mbp. In total, *HiRise* scaffolding joined 1,329 scaffolds. We then used the Assemblathon 2 script (https://github.com/ucdavis-bioinformatics/assemblathon2-analysis)[101] to assess assembly quality.

To annotate genes, we first masked repeats and low complexity DNA using *RepeatMasker* v4.1.1[102] using the 'Asteraceae' repeat database with Repbase database. After this first round, we ran *RepeatModeler* v2.0.1[103] on the masked genome to obtain a database of de novo elements. This database was subsequently used as input to *RepeatMasker* for a second round of masking the genome. To find gene models, we first assembled a transcriptome using PacBio HiFi data and following the IsoSeq3 pipeline (Pacific Biosciences). Processing of the RNA data involved clipping of sequencing barcodes (lima v2.0.0), removal of poly(A) tails and artificial concatemers (Isoseq3 refine v3.4.0), clustering of isoforms (Isoseq3 cluster v3.4.0), alignment of the reads to the reference genome using (pbmm2 align v1.4.0), characterisation and filtering of transcripts (SQANTI3 v1.0.0)[104]. Genome annotation was carried out using the *MAKER2* pipeline v2.31.9[105,106], using a combination of ab initio and homology-based gene predictions (using Asteraceae protein sets). Since no training gene models were available for *Scalesia atractyloides*, we used *CEGMA*[107] to train the ab initio gene prediction software *SNAP*[108]. In addition to the ab initio features, we used the IsoSeq transcriptome as a training set for the gene predictor *AUGUSTUS*[109], and as direct RNA evidence to *MAKER2*. Finally, when running *MAKER2* we specified, model_org = simle, softmask = 1, augustus_species = arabidopsis and specifying *snapphmm* to training of *SNAP*. To assess the quality of the gene models we used BUSCO and the viridiplantae odb v10 set[110–112]. Further checks of genome annotation quality were done using OrthoFinder with 4 other high-quality Asteraceae genomes (see below; Supplementary Fig. 02).

**Demographic reconstruction using PSMC.** To complement the *S. atractyloides* genome, we generated shotgun genomic data from DNA extracts of specimens of *S. helleri* B. L. Rob. and *S. stewartii* Riley, as well as the outgroup species *Pappobolus hypargyreus* and *P. juncosae*. Briefly, the *S. helleri* and *S. stewartii* specimens were extracted with a Qiagen DNeasy 96 Plant Kit, and the *P. hypargyreus* and *P. juncosae* extracts were previously reported (Fernández-Mazuecos et al.[9]). DNA extracts for these specimens were sent to the commercial provider Novogene for dsDNA library preparation, and they were sequenced on the Illumina NovaSeq platform in 150-bp PE mode. For these sequence data, we used *FastQC* v0.11.8 to check for quality of raw reads[113], identified adapters using *AdapterRemoval* v2.3.1, and removed them using *Trimmomatic* v0.39[114,115]. These sequences were then aligned to the *S. atractyloides* genome using the *mem* algorithm of *bwa*[116], and reads with a mapping quality below 30 were removed, resulting in a final sequencing depth of about ~15×. Alignments were then processed and analysed using *PSMC*[117]. Specifically, running *PSMC* involved calling variants using the *bcftools mpileup* and *call* algorithms, considering base and mapping qualities above 30 and read depths above 5[118], and posterior processing of the files using *fq2psmcfa*. For the *PSMC* run we specified a maximum of 25 iterations, initial theta ratio of 5, bootstrap, and a pattern of "4 + 25*2 + 4 + 6". To plot files we used the util *psmc_plot.pl* specifying a generation time of 3 years and a mutation rate of $6e^{-9}$, and constrained the y- and x axes to 50 and 20,000,000, respectively.

**Determination of subgenomes, and testing for subgenome dominance.** The determination of sugenomes involved two steps, the first using conserved portions of the genome (conserved orthologue sets or COS) and the second involving the exploration of *k*-mer distribution patterns. For the first step, we reasoned that homologous chromosomes would share COS. We used the Compositae-COS as baits (available through github.com/Smithsonian/Compositae-COS-workflow/raw/master/COS_probes_phyluce.fasta[119], running *phyluce* to mine for COS in the genome assembly[120,121]. This pipeline, however, is designed for single-copy COS, thus we manually modified the python script to return COS that are duplicated in the genome assembly. We then constructed a pairwise matrix of COS assignment using double-copy COS (Supplementary Information; Supplementary Table 03).

Duplicated COS provided a robust determination of chromosome pairings (homeolog chromosomes) but did not reveal which member of the pair belongs to which subgenome. To distinguish this, we performed a second step, where we analysed the *k*-mer spectrum[24]. We hypothesised that the period of separation of the two subgenomes led to the accumulation of different repeat content and transposable elements. To quantify *k*-mer abundance, we ran the software *Jellyfish*[122] for each chromosome independently, thus obtaining the per-chromosome frequency of *13-mers*. To ensure we targeted only repeats, we selected *13-mers* represented only >100 times represented at any given chromosome. To ensure that we targeted the period of separation (i.e. differential accumulation of TEs as hypothesised above), we compared *13-mer* frequencies in homeolog pairs and kept only *13-mers* that were at least twice as abundant within one member of each pair (e.g. if 500 counted in one member of the pair, then either <250 or >1000 counted in the other member). Using R, we computed a distance matrix and a hierarchical clustering, which neatly separated members of each pair into two groups (Supplementary Information; Supplementary Fig. 04). Finally, to confirm whether k-mers separated both subgenomes reliably, we repeated the distance matrix and hierarchical clustering analyses with a slight modification: we randomised chromosome pairs. Under a random pairing, we expected to obtain inconclusive results because chromosomes from the same subgenome should not have differentially represented TEs, and therefore subgenome groupings should not occur. Indeed, in line with this expectation, the randomisation of the chromosome pairs yielded inconclusive results.

To confirm the accuracy of subgenome assignment, we took two independent approaches. First, we created a Circos plot using the masked regions of the genome. To produce the Circos plot, we aligned the masked subgenomes to each other using *mummer*[123,124], and plotted the circos using the 'Circos, round is beautiful' software[125]. Second, we studied transposable element representation in each subgenome benefiting from the transposable element identification accomplished using *RepeatMasker*. Specifically, we obtained the list of different annotated transposable elements from *RepeatMasker* (e.g. RTE-BovB, LINE-L1, LINE-L2, Helitron, PIF-Harbinger Gypsy, Copia, CRE; Supplementary Data 4), and separated the families within these groups. For each family, we counted the number of elements present on each subgenome, and plotted all the families using raincloud plots[126]. To visualise genes and transposable elements along chromosomes, we used the R package Ideogram[127]. After identifying each subgenome, we ran BUSCO separately for each subgenome as a way of understanding subgenome-specific gene loss (Viridiplantae odb10 as specified above).

**Evolutionary history of the *Scalesia atractyloides* subgenomes and comparative genomics.** We searched the literature and NCBI for chromosome-level assemblies of the Asteraceae (February 5, 2021), downloading genomes assemblies of sunflower (*Helianthus annuus*[20]), the Canada fleabane (*Conyza canadensis*[128]), the 'mile-a-minute' weed (*Mikania micrantha*[129]), and lettuce (*Lactuca sativa*[21]). We downloaded the *Arabidopsis thaliana* genome from TAIR (Arabidopsis.org).

To obtain sets of orthologous genes, we ran OrthoFinder[130] on the predicted amino acid sequences (faa) and coding sequences (cds). Before running this software, we selected only the longest isoforms of both files, and removed sequences with stop codons as done by[131]. On the amino acid sequences we removed sequences with lengths below 30 bp using kinfin's filter_fastas_before_clustering.py script[132]. We ran OrthoFinder on various combinations of the genomes, including: (1) All Asteraceae, with subgenomes separated, (*S. atractyloides* subgenome A, *S. atractyloides* subgenome B, *C. canadensis*, *H. annuus*, *L. sativa*, *M. micrantha*); (2) All Asteraceae and the *Scalesia* genome (*S. atractyloides* (complete), *C. canadensis*, *H. annuus*, *L. sativa*, *M. micrantha*); (3) *A. thaliana* and subgenomes (*S. atractyloides* subgenome A, *S. atractyloides* subgenome B, *A. thaliana*). A representation of the run including all Asteraceae and the *Scalesia* genome and its processed results using an upset plot[133].

Dating of the speciation and the polyploidization event were performed independently. To date the speciation event, we obtained a phylogenetic tree, by running OrthoFinder with the two subgenomes separately (OrthoFinder run 1, above). OrthoFinder retrieved a tree of the single-copy orthologs, which was used for posterior analysis. This tree was converted as ultrametric using r8s[134]. To date the nodes of the tree, we converted branch lengths to time estimates using a calibration point of 6.14 Mya between *H. annuus* and *S. atractyloides* following recent literature[29], following the practice of recent literature[9].

Dating the polyploidization event was accomplished by combining the tree obtained by OrthoFinder (OrthoFinder run 1) and depended on transposable element distributions along the subgenomes as previously detailed by[24,25,135]. Briefly, this approach has a simple assumption: before the speciation event (which separates the ancestral lineages A and B) and after the polyploidization event (which brings the ancestral lineages A and B together), the accumulation of transposable elements will be similar on both subgenomes. In other words, transposable element families that are evenly represented on the subgenomes, therefore, represent the pre-speciation and post-allopolyploidization period. We focused on long-terminal repeats (LTRs) given their prevalence along the genome. To obtain high-quality LTR sequences, we started by using LTRharvest to identify LTR elements[136], followed by LTRdigest to process these elements. LTRdigest annotated features such as genes and domains inside LTRs, and helped refine the elements. To find features within the LTRs, we downloaded various PFAM domains provided in ref.[137], and complemented these by downloading and

concatenating them with the "Gypsy" and "Copia" domains from the PFAM online database. We converted the domains to HMMs using hmmconvert[138], and added HMMs from the Gypsy Database[139]. The identification and annotation of LTRs using these methods was done for the *S. atractyloides* and *Helianthus annuus* genomes, with the inclusion of the latter species serving as an outgroup for comparisons of genetic divergences. An important distinction relates to the LTR-element and the LTR-region: the LTR-element involves the whole transposable element including repeated regions and genes inside, while the LTR-region involves only the Long Terminal Repeat of the LTR-element. For the next analyses, we used only the LTR-region (as provided by LTR digest) as alignments were of better quality. Using LTR-domains of the *Scalesia* and *Helianthus* genomes as inputs, we ran OrthoFinder to obtain orthogroups consisting of closely related LTR-domains. We processed the OrthoFinder data by selecting orthogroups that met two assumptions: (1) they were in equal representation on both subgenomes (as hypothesised above); (2) and that were present in *Helianthus* (to calculate genetic distances, see below). Using orthogroups which met these two assumptions, we aligned orthogroups using mafft, and cleaned poorly aligned regions using Gblocks[140,141], with non-stringent options including 'allow smaller final blocks', 'allow gap positions within the final blocks', and 'allow less strict flanking regions'. After this, we further processed the data by removing sequences with more than 50% missing data, and re-checked whether numbers of TEs were still balanced between subgenomes, thus purging some further orthogroups. We then re-aligned the data using mafft and inferred a tree for each ortholog. We kept only orthogroups where the *S. atractyloides* LTR sequences were monophyletic, but where both subgenomes were non-monophyletic. For the final set of five orthogroups passing all this filtering (Supplementary Fig. 07), we calculated pairwise Jukes Cantor distances between each (1) *S. atractyloides* LTR-region, and between (2) *S. atractyloides* and *H. annuus*. The Jukes Cantor distances were plotted as frequency histograms in R (see Supplementary Fig. 06), and the peaks of the *Scalesia*-vs-*Scalesia* (golden on Supplementary Fig. 07) and *Scalesia*-vs-*Helianthus* (grey on Supplementary Fig. 07) were converted to million of years distance by a simple rule of three with the *Helianthus* divergence with *Scalesia* of 6.14 Mya (Supplementary Information; Supplementary Fig. 07).

**Signatures of selection and expanded gene regions**. Using the *Scalesia* genome together with the remaining Asteraceae genomes we ran CAFÉ analyses[142,143] to estimate significant gene-family expansions and contractions. Briefly, we did an all-by-all BLAST to identify orthologues in the dataset and estimated significantly expanded and contracted families using CAFÉ. To interpret the data we relied on Gene Ontology Annotation (GO). We obtained GOs for the annotated *Scalesia* genes by means of two complementary approaches: (1) by using the Interproscan command-line version[144], using the NCBI's Conserved Domains Database (CDD), Prediction of Coiled Coil Regions in Proteins (COILS), Protein Information Resource (PIRSF), PRINTS, PFAM, ProDom, ProSitePatterns and ProSiteProfiles, the Structure–Function Linkage Database (SFLD), Simple Modular Architecture Research Tool (SMART), SUPERFAMILY, and TIGRFAMs databases; (2) by extracting the curated Swiss Prot database from UniProt (Viridiplantae) and blasting the *Scalesia* genes to this database, keeping hits with an *e*-value below 1e-10. We then extracted the GOs from genes from the database and assigned these to *Scalesia*'s correspondent orthologs. Genes belonging to significantly expanded gene families in the *S. atractyloides* genome were analysed using a GO enrichment analysis. To do so, we used the TopGO package using the 'elim' algorithm which takes GO hierarchy into account[145,146], this was then summarised using *REVIGO*[147].

To test which genes are under positive selection in *S. atractyloides* genome, we retrieved the orthogroups from all Asteraceae, and aligned the cds from each orthogroup using prank[148]. Considering the divergence in the genomes, as well as evidence for fast evolution in Asteraceae genomes (including this paper), we ran zorro[149], to assess the alignments. Zorro scores each alignment position between 0 and 10, and we selected only alignments with an average score position of 5 or greater. For each of these, we inferred a tree using IQtree and ran HyPhy using its aBSREL positive selection test[150,151]. To summarise these results we: (1) ran a GO enrichment analysis (as specified above) to obtain general insights, plotting results using *REVIGO*; (2) identified the *Arabidopsis* ortholog to each of the *Scalesia* genes under selection using BLAST, and analysed the *Arabidopsis* literature for that particular gene (Supplementary Information; Supplementary Table 08); (3) we ran a STRING analysis (Supplementary Fig. 07) using the *Arabidopsis* orthologs[152], thereby exploring the potential protein-protein interactions among genes under selection. Interaction scores of edges were calculated based on the parameters Experiments, Co-expression, Neighborhood, Gene fusion and Co-occurrence. Edges with interaction score higher than 0.400 were kept in the network. After excluding genes with no physical connection, the STRING network had 627 nodes with 470 edges (PPI enrichment *P* value < 0.001). To simplify the densely connected network into potential biologically functional clusters, we used the distance matrix obtained from the STRING global scores as the input to perform a k-means clustering analysis (number of clusters = 6). Four of the six clusters are enriched for biological processes related GO terms. Cluster 1 (red bubbles) were enriched for the GO term metabolic processes, cluster 3 (lime green bubbles) for histone modifications and chromosome organisation, cluster 4 (green bubbles) for response to light, and cluster 6 (purple bubbles) for ribosomal large subunit biogenesis and RNA processing.

**Reporting summary**. Further information on research design is available in the Nature Research Reporting Summary linked to this article.

## Data availability

The raw data generated in this study have been deposited in the ENA database under accession PRJEB52418. The assembly and the annotation files are available at Cerca, J. (2022), *Scalesia atractyloides* genome assembly, Dryad, Dataset, https://doi.org/10.5061/dryad.8gtht76rh.

## Code availability

An overview of the bioinformatic methods is provided in https://github.com/jcerca/Papers/tree/main/scalesia_genome.

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

## Acknowledgements

J.Ce. is grateful to Simen R. Sandve for fruitful discussion and Martin LaForest for sharing genome annotations. We thank Henning Adsersen for the botanical expertise and logistical support that enabled use of the University of Copenhagen botanical collections. Jennifer Mandel kindly shared the Asteraceae COS. The collection and photography of specimens, and the preparation of this manuscript, benefited enormously from the cooperative assistance of the personnel of the Charles Darwin Foundation Research Station, who made arrangements for collecting trips, arranged laboratory space, and offered encouragement and support throughout the project. *Scalesia* specimens were initially collected under Galápagos National Park research permit number PC-001/98 PNG and were further normalised via Ecuador Ministry of the Environment genetic permit number MAAE-DBI-CM-2021-0213. This publication is contribution number 2426 of the Charles Darwin Foundation for the Galápagos Islands. This work was supported by the Norwegian Research Council via project number 287327 awarded to M.D.M., and a travel grant (project number 287327) granted to J.Ce. and M.D.M.

## Author contributions

J.Ce. designed the experiment, processed and analysed the data and drafted the manuscript. B.P., J.M.L.G., A.R.-C., J.Ca., Q.L., S.B., J.V., S.L. and D.M. helped J.Ce. analyse the data. J.L. was responsible for the flow cytometry analyses. C.K. and L.S.-B. helped in retrieving DNA/RNA, N.W., M.N., P.J.D. and G.R.-T. obtained permits. M.F.-M., P.V. and R.M. obtained the outgroups. G.P., A.S., N.S., N.N., O.S., T. G., J.H. L.-M., L.R. and L.N. contributed expertise in data generation and interpretation, A.S. assisted with the TE analyses. M.D.M. obtained funding, generated the data and helped interpreting. Every author commented, revised and approved the manuscript.

## Funding

## Competing interests

The authors declare no competing interests.

## Additional information

[1]Department of Natural History, NTNU University Museum, Norwegian University of Science and Technology, Trondheim, Norway. [2]Centre for Evolutionary Hologenomics, The GLOBE Institute, Faculty of Health and Medical Sciences, University of Copenhagen, Øster Farimagsgade 5, 1353 Copenhagen, Denmark. [3]Centre of Excellence for Omics-Driven Computational Biodiscovery, Faculty of Applied Sciences, AIMST University, Kedah, Malaysia. [4]Department of Botany and Biodiversity Research Centre, University of British Columbia, Vancouver, BC V6T 1Z4, Canada. [5]Department of Evolution, Ecology, and Behavior, University of Illinois at Urbana-Champaign, Champaign, IL, USA. [6]Department of Chemistry, Biotechnology and Food Science, Norwegian University of Life Sciences, Ås, Norway. [7]Natural History Museum, University of Oslo, Oslo, Norway. [8]Villum Centre for Biodiversity Genomics, Section for Ecology and Evolution, Department of Biology, University of Copenhagen, Universitetsparken 15, 2100 Copenhagen, Denmark. [9]Department of Plant Biology, University of California, Davis, Davis, CA 95616, USA. [10]Centre for Functional Ecology, Department of Life Sciences, University of Coimbra, Calçada Martim de Freitas, 3000-095 Coimbra, Portugal. [11]Department of Geosciences and Natural Resource Management, University of Copenhagen, Rolighedsvej 23, 1958 Frederiksberg C, Denmark. [12]Estación Científica Charles Darwin, Fundación Charles Darwin, Santa Cruz, Galápagos, Ecuador. [13]Department of Botany and Plant Physiology, University of Malaga, Malaga, Spain. [14]Colegio de Ciencias Biológicas y Ambientales COCIBA & Extensión Galápagos, Universidad San Francisco de Quito USFQ, Quito 170901, Ecuador. [15]Galapagos Science Center, USFQ, UNC Chapel Hill, San Cristobal, Galapagos, Ecuador. [16]Estación de Biodiversidad Tiputini, Colegio de Ciencias Biológicas y Ambientales, Universidad San Francisco de Quito USFQ, Quito, Ecuador. [17]Courtesy Faculty, Department of Wildlife Ecology and Conservation, University of Florida, 110 Newins-Ziegler Hall, Gainesville, FL 32611, USA. [18]Departamento de Biología, Universidad Autónoma de Madrid, 28049 Madrid, Spain. [19]Departamento de Biodiversidad y Conservación, Real Jardín Botánico (RJB-CSIC), Plaza de Murillo 2, 28014 Madrid, Spain. [20]Department of Biology, Fort Lewis College, Durango, CO 81301, USA. [21]Department of Ecology, Environment and Plant Sciences, Stockholm University, SE-106 91 Stockholm, Sweden. [22]Department of Archaeology, University of York, York, UK. [23]Department of Biotechnology and Food Science, Norwegian University of Science and Technology, Trondheim 7491, Norway. [24]School of Biological Sciences, University of East Anglia, Norwich Research Park, NR4 7TU Norwich, UK. [25]Department of Organismal Biology, Evolutionary Biology Centre (EBC), Science for Life Laboratory, Uppsala University, 75236 Uppsala, Sweden. [26]The Natural History Museum of Denmark, University of Copenhagen, Copenhagen, Denmark. [27]Department of Plant Biology, University of Georgia, Athens, GA 30602, USA. ✉email: jose.cerca@gmail.com; mike.martin@ntnu.no

