## [Peer Review File · Nature Communications]

The genomic basis of the plant island syndrome in Darwin's giant daisiesReviewers' Comments:

Reviewer #1:

Remarks to the Author:

The manuscript of Cerca et al. has reported a genome of the endangered *Scalesia atractyloides* and tries to interpret the genome from an exciting point of view, i.e., the genomic basis for the plant island syndrome. I have great interest in reading the manuscript, which has confirmed the allotetraploid origin of *Scalesia*, identified structural variations between the two subgenomes, and found some genes that may be under positive selection via sequences or gene family sizes. However, after reading the manuscript, I feel that studying the plant island syndrome requires detailed comparisons between the insular species and their close relatives on the continent (if they exist). Although I can understand that this genome can serve as a reference genome for future studies, some of the comparative analyses are neither fairly sound nor entirely in support of the conclusions in the abstract.

1.The introduction is a bit confusing. In the first read, it was fascinating and enlightening. However, after reading the whole manuscript, I found it mixed the concepts of island syndrome and adaptive radiation. Although the title highlights "the plant island syndrome," the introduction hardly mentions specific phenotypes in *Scalesia* species compared to their continental counterparts, such as *Scalesia* species becoming trees or shrubs on islands, which are very rare in Asteraceae. In contrast, the introduction includes many aspects of adaptive radiation for the *Scalesia* species, like their distributions and habitats. If it is a paper about adaptive radiation, I would expect more sequenced *Scalesia* species, mimicking the work on Darwin's finches by Lamichhaney et al. (2015). Suppose the introduction does not sufficiently explain the specific phenotypes of the plant island syndrome in *Scalesia* (better with some photos as most people are not familiar with the species). In that case, it is difficult to understand the examples of genes under selection.

2.Line 165: To show that the proportion of repetitive elements and the number of genes are within the variation of sequenced Asteraceae genomes, only one example of the sunflower is not enough because it cannot show the variations.

3.Line 204-207: After identifying the two sets of subgenomes, the authors used some TE families to verify the results. However, it seems circular reasoning for me because they first found TE families that were unevenly distributed in the two subgenomes. Then they claimed these unevenly distributed TE families support the classification of the two subgenomes. Also, Figure 2B only shows six TE families, which is unexpected for a study at the genomic level.

4.Line 217-223: The methods for dating the divergence between the two parental lines and the event of tetraploidization are not fully clear to me. First, the only calibration in the analysis is a point at 6.14 mya for the divergence between sunflower and *Scalesia*. At least allowing some variations for the divergence may make the dating more promising. Second, in Suppl. Figure 4 (which has two figure legends), only five LTR orthogroups were used to date the tetraploidization event, which is again unexpected for a genome-level study. Also, the dates on the figures are out of the explanation. A common way to date hybridization events with genomic data is to depict TE proportions against TE divergence (JC distance is acceptable here) as described in, for instance, the paper of the common carp genome by Xu et al. (2019).

5.Line 246-249: The *Scalesia* genome has even distributions of TEs along chromosomes, similar to bryophyte genomes and the sunflower genome. However, these observations have been only made by genomes published recently, and centromeres and telomeres are still notorious for genome assembly. Hence, I wondered if there is other evidence to support the even distribution of TEs in these species.

6.Line 259-261 and Figure 3B-E: Strangely, the authors identified fewer genes and fewer pseudogenes in subgenome B than subgenome A. Would this suggest that subgenome B already lost quite some genes before the two parental lines hybridized? Or subgenome A may have more small-scale

duplications recently, which may produce pseudogenes? Also, the distributions in Figure 3B-E (and Figure 2B) overtake the box plots and data points. Therefore, I would suggest only using box plots and the data points, as each subgenome only has 17 data points.

7.Line 267-271: The claims here sound contradicted or at least unclear. The authors claim that there has been no drastic rearrangement of subgenomes. Still, they fail to clarify with evidence whether they are talking about rearrangements before or after the tetraploidization event. If it is the latter, the final sentence seems to have no support.

8.Line 284-307 and Figure 4: The authors studied rearrangements between the two subgenomes. However, the results look a bit strange to me. In Figure 4A, the same color does not always code for homologous chromosome pairs. For instance, chr15 in subgenome A and chr13 (Figure 2A) are different colors. Also, chr15 is the second chromosome in subgenome A, and chromosome 13 is the third in subgenome B. This makes me think about how the orders of chromosomes were determined because the order seems to relate to my second question here. According to the authors, 70% of the studied genes were translocated, but Figure 4B and Figure 2A seem to indicate the opposite. Although visualization might mislead sometimes, more detailed results would be helpful here.

9.Line 342: The authors have identified 920 genes under selection, which to me are all very important if they are really under selection. However, despite performing GO enrichment analysis, they randomly selected 100 genes in detail to look at. This is strange because I do not think the 100 genes can represent the 920 genes. This is also not reproducible if a reader wants to select 100 genes by him/herself. In addition, finding a certain number of genes involved in photosynthesis, leaf morphology, and stress responses are probably expected even when someone randomly selects 100 genes from all the genes in the genome. Therefore, I am not entirely convinced by the current analysis here.

Minor issues:

1.Figure 2a. Some chromosome IDs are different from the rest, such as 116, 1632, 1634, and 1633. You may consider renaming those or explaining the naming rules in the figure legends.

2.Line 273: "Fast evolutionary rates" sounds ambiguous. It is not very clear to know what the authors want to say here.

3.The authors mention "Supplementary information" several times throughout the manuscript, but I could not find the document in the review materials.

Reviewer #2:

Remarks to the Author:

The authors assembled the genome of the Galapagos endemic, allotetraploid daisy species *Scalesia atractyloides* to identify the genomic basis of the island syndrome and the links between genome evolution and adaptive radiations on islands.

While the goals of the paper are interesting and the performed genome assembly and subsequent analyses generally solid, I think that the links between genomic characteristics and island radiations remain very speculative. A considerable part of the analyses is devoted to characterizing the two subgenomes of the allotetraploid species, and this is the stronger part of the analyses. However, the proposed links between genomic characteristics and the rapid, pronounced phenotypic changes associated with island life are tenuous and unsatisfactory. Additionally, I feel that the paper suffers from some degree of conceptual confusion about fundamental topics of island biology. For example, the paper remains vague about the concepts of island radiation vs. insular adaptive radiation, using the terms interchangeably, even though they are not. I will justify my assessment in greater detail

below.

L91: The concept of island syndrome must be better differentiated from the scenario of colonization of other ecological niches that are not on islands. In other words, novel selective pressure likely arise after colonization of any empty ecological space, whether it is on islands or not.

L102-103: Explain how plants change as a result of island colonization, defining directionality of character change from mainland to island.

L112 (entire paragraph): The authors need to make a clear distinction between radiations and adaptive radiations. Not all radiations are adaptive. In adaptive radiations, diversification is driven by adaptation (for example to different habitats), rather than by allopatric speciation followed by adaptation and phenotypic divergence. This is an important conceptual distinction that does not come up in the paper. The authors seem to merge the concepts of radiations and adaptive radiations. A lot has been written on this distinction, starting from the classic book by Givnish and Sytsma on the topic of island radiations.

For example, in the studied group, the hypothesis of adaptive radiation would receive at least some degree of initial support if species occurring in different habitats on the same island evolved from a single common ancestor. Another way to put it is that the hypothesis of adaptive radiation would receive some tentative support if species co-occurring on the same island in different habitats were more closely related to each other than species occurring in the same (or similar) habitats on different islands. The authors should discuss these concepts and explain whether the adaptive nature of the studied radiation has been proven, for example, using the abovementioned phylogenetic approach. As the famous evolutionary biologist GC Williams noted already in 1966, the concept of adaptation is an onerous one, which is difficult to prove, so it is important that it is used very carefully and only if some analytical evidence is available.

The fact that this group of island plants presents variation of morphological features is a necessary but not sufficient condition to prove the adaptive nature of the radiation.

Additionally, the concept of radiation itself is often linked to the speed of diversification, so the authors should also explain whether previous evidence proves this point, typically through some sort of molecular dating analysis (or other means, if available).

If no analytical evidence is available to prove that the mentioned plant diversification is indeed an adaptive radiation, then the authors should refrain from using this term.

L148-149: It seems like this is not an especially complete genome assembly. What is the percentage of the genome that was actually assembled?

L208: Define what paleoallopolyploids are and the available evidence that the studied species is indeed one of them.

L315-316: Thus, the hypothesis of genome miniaturization potentially connected with island colonization in *Scalesia* is unwarranted, since the direction of change has not been established. Even if this were the case, what would the adaptive value of genome miniaturization on islands be? If the authors wanted to establish a clear link between this genomic feature and adaptive island radiation, they would need to be much more specific about a potential mechanism.

L351-352: This evidence on leaf morphology is very tenuous.

L366-368: OK, but how is this linked to island life and island radiations?

L371-372: But again, this would not be specific to dry conditions just on islands.

L381-382: I feel that the authors have not established a solid link between the characteristics of the target plant genome and island adaptive radiation. For example, they list a series of genes under selection with very generic functions, e.g. genes associated with growth and transitions between life stages. Are these genes under selection also in non-insular relatives of the studied species? It seems to me this kind of comparative genomic analyses of genes under selection in island plants versus non-insular relatives would be necessary to make a tentative case for the identification of links between

genomic characteristics and adaptations to island life and the proposed adaptive radiation of this groups of plants.

L395-396: Very speculative and vague.

I found a few spelling or similar mistakes:

L78: specular instead of spectacular

L342: misspelled revolution for evolution

Figure 4 legend seems to have a lingering editing mistake.

Reviewer #3:

Remarks to the Author:

This manuscript provides the most detailed assessment of genomic evolution in a plant lineage resulting from adaptive radiation in a remote archipelago. Several of the findings are noteworthy, especially considering the widespread incidence of ancestral, recent allopolyploidy in flowering plant lineages that have diversified extensively in oceanic islands. Beyond demonstrating a recent allopolyploid origin for *Scalesia*, which is unquestionably the most important example of adaptive radiation in the Galápagos flora, the findings of a lack of subgenome dominance in retention and expression of genes, considerable structural differentiation of the two subgenomes (by translocations and inversions) and high gene density within inverted regions, and characterization of many of the genes under positive selection as ones previously associated with functions that arguably may have been important in adaptive evolution are all major contributions of this study. In addition, the work is methodologically significant for using fossil transposable elements (TEs) to assign genes to subgenomes, which is an important challenge in polyploid plant genomics. The methodologies in general are sound, cutting-edge, and detailed sufficiently for replication. The results of this study will be of value to plant evolutionary biology in general, including phytogeography and plant evolutionary genomics – it is of broad interest. The analyses are sound and sufficient to back up the conclusions presented. The only minor concern worth mentioning here is that phylogenetic divergence time estimates should be indicated as approximate (using ca. or ~) unless error in the estimates are provided.

Below are suggested minor edits, indicated by line number on the submitted pdf of the manuscript

Line 66: Change “change in ploidy” to “ancestral polyploidy”? As noted later in the manuscript, change in ploidy within remote archipelagos is rare and not truly part of the island syndrome – but polyploidy is associated with successful colonization and radiation, as a pre-condition rather than polyploidization after colonization (as in *Scalesia*)

78: spectacular (spelling)

108: Need reference(s) for Juan Fernandez radiations

115: Change “littoral” to “littoral zone” and “arid” to “arid zone” to avoid possible confusion that these are forest types rather than ecological zones

119: capitulum (spelling) AND change “The outstanding” to “This outstanding”

123: Change “for other island floras” to “for several island floras” (including Galápagos)

126: Change “most basal” to “one of the deepest” (that is, there are two deepest (sister) lineages in any clade, unless the base is unresolved)?

207: Change "of allopolyploid lineage" to either "of allopolyploid origin" or "an allopolyploid lineage")

258: Add "in" before "paleopolyploid plant genomes"

270 through 271: This same speculation about the importance of allopolyploidization in adaptation to insular environments was proposed by Barrier et al. (1999) Mol. Biol. Evol. upon discovery that the silversword alliance genome is allotetraploid

301: Replace commas with decimal points in 693,2 and 586,5

366 through 367: Does "absence of permanent light conditions" mean partially shaded conditions (meaning unclear)?

396: Change "encounter" to "encountered"

423: Change "flowering of plants" to "flowering plants"

424: Change "first and solid" to "first solid"

461: Change "chromatic" to "chromatin"

461 through 464: Change present-tense to past-tense for description of methods here

489: Change "detecting a removing " to "detecting and removing"

490: Delete "and" after "input to"

492: Delete "was reduced"

561: Change "which" to "that"

570: Change "In specific" to "Specifically"

581 through 583: Best to delete "the" before common names of the four species of Asteraceae (i.e., change "the lettuce" to "lettuce").

595 through 596: The sentence spanning these lines is incomplete

606: Change "which" to "that"

615: Change "insides" to "inside"

617: Change "which" to "that"

620: Change "not stringent" to "non-stringent"

628: A word is missing after "H. annuus" at end of sentence

650: Change "were" to "was"

661: Change "gene" to "genes"

670 through 671: Change to "Four of the six clusters ..." (that is, to avoid beginning sentence with a numeral)

Below we present a point-by-point response to the concerns raised by the three reviewers. In brief, we have:

- 1 - Substantially revised the abstract and the introduction, to clarify any conflicts or miscommunications regarding the concepts of 'adaptive radiation' and 'island syndrome';
- 2 - Removed several points (and even two entire paragraphs) following the reviewers' recommendations regarding speculation based on too little data. These have not changed the overall content of the manuscript, since we only removed minor points (e.g. reference to island taxa having novel enemies);
- 3 - Cleared up the majority of minor issues highlighted by the reviewers, including altering the labels of our chromosomes

We believe that addressing these issues has substantially improved our manuscript, and thus we have added a note of appreciation dedicated to the reviewers to the Acknowledgements section.

José & Mike

REVIEWER COMMENTS

Reviewer #1 (Remarks to the Author):

#R1C1 -The manuscript of Cerca et al. has reported a genome of the endangered *Scalesia atractyloides* and tries to interpret the genome from an exciting point of view, i.e., the genomic basis for the plant island syndrome. I have great interest in reading the manuscript, which has confirmed the allotetraploid origin of *Scalesia*, identified structural variations between the two subgenomes, and found some genes that may be under positive selection via sequences or gene family sizes. However, after reading the manuscript, I feel that studying the plant island syndrome requires detailed comparisons between the insular species and their close relatives on the continent (if they exist). Although I can understand that this genome can serve as a reference genome for future studies, some of the comparative analyses are neither fairly sound nor entirely in support of the conclusions in the abstract.

We are grateful for the reviewer's assessment. As the reviewer points out, the *Scalesia* system is interesting in that it represents an excellent system to understand ploidy in island plant radiations, which are often eclipsed by Darwin's finches and other more 'charismatic' groups. The identification and magnitude of structural variants between subgenomes is novel (first reported by us), and uses cutting-edge analysis. Below we address the #R1's concerns.

#R1C2 - The introduction is a bit confusing. In the first read, it was fascinating and enlightening. However, after reading the whole manuscript, I found it mixed the concepts of island syndrome and adaptive radiation. Although the title highlights "the plant island syndrome," the introduction hardly mentions specific phenotypes in *Scalesia* species

compared to their continental counterparts, such as *Scaevola* species becoming trees or shrubs on islands, which are very rare in Asteraceae. In contrast, the introduction includes many aspects of adaptive radiation for the *Scaevola* species, like their distributions and habitats. If it is a paper about adaptive radiation, I would expect more sequenced *Scaevola* species, mimicking the work on Darwin's finches by Lamichhaney et al. (2015). Suppose the introduction does not sufficiently explain the specific phenotypes of the plant island syndrome in *Scaevola* (better with some photos as most people are not familiar with the species). In that case, it is difficult to understand the examples of genes under selection.

The reviewer is absolutely right in that the relevant concepts should have been made more clear in the Introduction (and abstract). We have substantially revised parts of the Introduction and abstract to shift the focus and context from adaptive radiation to the island syndrome. We appreciate this suggestion, which has made our work much more clear.

All the additions in this review are presented in bold, italics, underlined

On the abstract:

“Oceanic archipelagos comprise multiple disparate environments over small geographic areas and are isolated from other biotas. These conditions have led to some of the most **rapid and spectacular phenotypic changes, which are often repeated, thus offering a unique chance to characterise its genomic basis.** **These** repeated patterns of evolutionary change in plants on oceanic archipelagos, **or the** ‘plant island syndrome’, include changes in leaf **phenotypes**, acquisition of perennial life-style, flowering period and self-compatibility, and ancestral ploidy.”

In the introduction it reads:

“Organisms colonising these regions encounter highly distinct microenvironments that provide abundant ecological niches and thus ideal conditions for **rapid and pronounced phenotypic change** (Lomolino, Riddle, and Whittaker 2017).”

“Because the most prominent examples of **island syndromes feature animal lineages**, our understanding of these phenomena in plant lineages lags (Burns 2019). As plants colonise archipelagos, they typically and repeatedly undergo directional shifts in leaf morphology, dispersal ability, lifespan and size (Burns 2019). **This is well exemplified by the iconic, yet understudied, daisies in the genus *Scaevola*** (Blaschke and Sanders 2009; Fernández-Mazuecos et al. 2020; Crawford et al. 2009; U. Eliasson and U 1974). This group consists of ca. 15 species, which have colonised moist forests, littoral zones, arid zones, dry forests, volcanic soils, lava gravels and fissured environments across varied elevations (Itow 1995; Blaschke and Sanders 2009). The morphological changes undergone by *Scaevola* include an increased woodification, leaf-morphology variation, simplified inflorescences, increased growth rates, and gigantism - as expected by the plant island syndrome. Indeed, this outstanding phenotypic and ecological variation has led authors to refer to this group as the ‘Darwin finches of the plant world’ (Stöcklin 2009). All *Scaevola* species are ancestrally tetraploid ($2n=4x=68$) (Ono 1967; Uno Eliasson 1974), and the polyploid genetics may have provided the genetic grist for the diversification, as has been speculated for several island floras (Meudt et al. 2021).”

#R1C3 - Line 165: To show that the proportion of repetitive elements and the number of genes are within the variation of sequenced Asteraceae genomes, only one example of the sunflower is not enough because it cannot show the variations.

The reviewer is completely right. We have now added more comparisons. Now it reads:

“The proportion of annotated repeats and number of genes is within the variation reported for Asteraceae. **As an example**, the assembly of the closely related sunflower (*Helianthus annuus*) reference genome includes ~52,000 protein coding genes and has a repeat content of 74% (Badouin et al. 2017), **the lettuce genome includes 74 % repeats and ~39,000 protein coding genes (Reyes-Chin-Wo et al. 2017), and the Hawaiian *Bidens* genome has 70-74 % repeats (Bellinger et al. 2022).**”

#R1C4 - Line 204-207: After identifying the two sets of subgenomes, the authors used some TE families to verify the results. However, it seems circular reasoning for me because they first found TE families that were unevenly distributed in the two subgenomes. Then they claimed these unevenly distributed TE families support the classification of the two subgenomes. Also, Figure 2B only shows six TE families, which is unexpected for a study at the genomic level.

We believe there has been a misunderstanding due to the unclear writing in our previous text. To clarify, we used two methods to identify the subgenomes:

1. We used a set of conserved orthologs to identify homeolog chromosomes, although at this point their affinity to particular subgenomes was not yet defined;
2. Then, we used transposable elements (hereafter TEs) to assign the members of each pair to either subgenome A or subgenome B.

Since the assignment of the transposable elements is a relatively complex method, we obtained some of the most unevenly represented transposable element families and displayed them on Figure 2B as a proof of concept. In other words, by showing differently represented families, we demonstrate that TEs are indeed able to separate subgenomes, as these families were active during the separation of the two ancestral species. Figure 2B is therefore intended to show the rationale, utility, and power of TEs in the method, and thus, we have included only 6 TE comparisons to represent, but these are not, in any way, the only differentially represented TE families.

We have changed the results section so that it reads better and, in particular, does not invoke circular reasoning. Now it reads:

“To **show the power** of this assignment, **and, in particular, the occurrence of differentially represented transposable element families** we explored the output from RepeatMasker **by plotting** transposable element families unevenly represented across subgenomes (Figure 2B).

#R1C5 Line 217-223: The methods for dating the divergence between the two parental lines and the event of tetraploidization are not fully clear to me. First, the only calibration in the

analysis is a point at 6.14 mya for the divergence between sunflower and *Scalesia*. At least allowing some variations for the divergence may make the dating more promising. Second, in Suppl. Figure 4 (which has two figure legends), only five LTR orthogroups were used to date the tetraploidization event, which is again unexpected for a genome-level study. Also, the dates on the figures are out of the explanation. A common way to date hybridization events with genomic data is to depict TE proportions against TE divergence (JC distance is acceptable here) as described in, for instance, the paper of the common carp genome by Xu et al. (2019).

We apologise for the lack of clarity. This is better explained in the methods below.

The major issue with dating approaches is that different genomic regions will have different mutation rates. In the context of an island radiation, the evolutionary history of organisms is characterised by bottlenecks and founder effects, and fast population growth associated with the appearance of new resources/areas (e.g. new islands; change of climate as a result of novel islands emerging). In this way, using a whole-genome approach to date splits is challenging. In the light of these challenges, we used the robust dating calibrations provided by Fernandez-Mazuecos (included in this work), which were published in 2020, rather than repeating their analysis.

We added:

“Dating of this tree was done by constraining the node separating the *Scalesia* subgenomes and *Helianthus* as 6.14 Mya (Figure 2C), following recent literature (Mandel et al. 2019; Fernández-Mazuecos et al. 2020).”

The reviewer is right that we used Jukes Cantor distances to depict evolutionary divergence in TEs. This has now been clarified in the methods section.

#R1C6 Line 246-249: The *Scalesia* genome has even distributions of TEs along chromosomes, similar to bryophyte genomes and the sunflower genome. However, these observations have been only made by genomes published recently, and centromeres and telomeres are still notorious for genome assembly. Hence, I wondered if there is other evidence to support the even distribution of TEs in these species.

The even distribution of TEs along chromosomes was noticed by one of the senior co-authors who is well versed in plant genome assembly. We agree with the reviewer that further genomes will further elucidate patterns of TE-accumulation along the genome, especially as the taxonomic breadth of available plant genomes is increased. We have added a sentence saying that we need more genomes. Now, it reads:

“Even distributions of transposable elements were also observed in the sunflower genome (Badouin et al. 2017), which may be indicative of particular transposable element regulation in the Heliantheae. This pattern should be confirmed as more Heliantheae genomes are sequenced.”

#R1C7 Line 259-261 and Figure 3B-E: Strangely, the authors identified fewer genes and fewer pseudogenes in subgenome B than subgenome A. Would this suggest that subgenome B already lost quite some genes before the two parental lines hybridized? Or

subgenome A may have more small-scale duplications recently, which may produce pseudogenes? Also, the distributions in Figure 3B-E (and Figure 2B) overtake the box plots and data points. Therefore, I would suggest only using box plots and the data points, as each subgenome only has 17 data points.

With respect to the lower number of pseudogenes and genes in subgenome B. Several factors may be in play:

1. It could be that the loss of genes is accelerated in subgenome B. In this case, and considering the high % and number of TEs in the genome, it may be that after 'pseudogenification', the pseudogene-tracks are quickly lost by TEs jumping in and out.
2. It could also mean that subgenome B has had a lower number of genes, as the reviewer points out.
3. It could also mean a part of subgenome B fractionated and joined a chromosome that was identified as subgenome A.
4. It could also mean that subgenome A may have more small-scale duplications recently, as the reviewer points out. But we would not expect uneven duplications along the genome.

However, because all of this is speculative, and warrants further work, we refrain from speculating in the absence of evidence.

We have now updated the figures, removing the histogram as the reviewer suggest. For instance, Figure 3 B-E look the following way:

#R1C8 - Line 267-271: The claims here sound contradicted or at least unclear. The authors claim that there has been no drastic rearrangement of subgenomes. Still, they fail to clarify with evidence whether they are talking about rearrangements before or after the tetraploidization event. If it is the latter, the final sentence seems to have no support.

We do not think that there has been a substantial rearrangement of the subgenomes after polyploidization and as predicted by the diploidization hypothesis. However, logically, if there are no drastic rearrangements of either subgenome, this must also have been true for the brief period of divergence before the tetraploidization event, as any drastic rearrangements during that period would have been inherited in the polyploid lineage.

Now, this section of the text reads:

“This indicates that during the past ~3.76 million years, during which the two subgenomes have been unified in the same organism, there has not been a drastic rearrangement of either subgenome *relative to the other*, despite a smaller accumulation of genes and pseudogenes on subgenome A. ***This suggests diploidization is slowed down and,*** o explain this, we speculate that *Scalesia*’s adaptation to insular environments has benefitted from the genetic variation and diversity stemming from the allopolyploidization event (Barrier et al. 1999).”

#R1C9 - Line 284-307 and Figure 4: The authors studied rearrangements between the two subgenomes. However, the results look a bit strange to me. In Figure 4A, the same color does not always code for homologous chromosome pairs. For instance, chr15 in subgenome A and chr13 (Figure 2A) are different colors. Also, chr15 is the second chromosome in subgenome A, and chromosome 13 is the third in subgenome B. This makes me think about how the orders of chromosomes were determined because the order seems to relate to my second question here. According to the authors, 70% of the studied genes were translocated, but Figure 4B and Figure 2A seem to indicate the opposite. Although visualization might mislead sometimes, more detailed results would be helpful here.

Figure 4A aims to display the correspondence between *Helianthus* and subgenomes. In this figure, subgenomes A and B are ordered independently because they are separately paired to the center genome, which is *Helianthus*. In other words, the ordering (and color) in panel A is not prioritizing the subgenome to subgenome pairing like in 4B. It prioritizes the pairing of each subgenome independently to the *Helianthus* assembly. In panel B, the chromosomes are also ordered by size, therefore in subgenome B 13 is the third one, since its the third largest.

The reviewer is totally right that the image did not match the text. We noticed an error with our analysis, and have now corrected it. In specific, now in the text, we include clusters only with 5 or more genes (to diminish the noise-to-signal ratio). Correction of this error did not change the main interpretation of our results. Now it reads:

“Comparing the two *Scalesia* subgenomes, we found 4,379 FP genes (comprising 111 clusters of 5 or more genes), 5,642 IP genes (78 clusters), 747 FT genes (31 clusters), and 1,488 IT genes (18 clusters), totalling 12,256 genes included in the analysis (Figure 4B). Thus, the majority of the genes have been inverted (7130 genes. In terms of genome length, we classified 1.45 Gbp as FP, 1.15 Gbp as IP, 346.4 Mbp as FT, and 343.3 Mbp as IT. Despite the fact that we were able to identify homeologs and the subgenomes, the synteny plots confirms there are rapid rates of chromosomal

rearrangements in the Asteraceae (Badouin et al. 2017), and suggest a central role of inversions in the family."

#R1C10 - Line 342: The authors have identified 920 genes under selection, which to me are all very important if they are really under selection. However, despite performing GO enrichment analysis, they randomly selected 100 genes in detail to look at. This is strange because I do not think the 100 genes can represent the 920 genes. This is also not reproducible if a reader wants to select 100 genes by him/herself. In addition, finding a certain number of genes involved in photosynthesis, leaf morphology, and stress responses are probably expected even when someone randomly selects 100 genes from all the genes in the genome. Therefore, I am not entirely convinced by the current analysis here.

We agree that performing an in-depth analysis of the entire set of 920 genes would lead to better results. However, there are several things to take into consideration here:

1. The GO enrichment analysis confirmed the general pattern revealed by the in-depth investigation of the set of 100 random genes under selection. Thus the in-depth investigation of this gene set does not stand alone and should be considered as a complement to the GO enrichment analysis. This allowed us to efficiently classify and confirm the patterns from the 100 gene classification.
2. Analysing gene-by-gene took an extraordinary amount of time and effort. The process involved looking for the gene on Arabidopsis.org, selecting papers associated with that gene, and digesting them. This involved examining more than 150 papers, a process that is detailed in the supplementary materials.
3. An in-depth analysis and discussion of the remaining 820 genes would make the paper substantially more verbose.
4. Fine-grain analysis of the literature, like we did for 100 genes, are extremely rare in the field of comparative genomics. We have done it for a portion of the genes involved and we consider it a plus.

Finally, we disagree that the results are not reproducible. We provide the gene list as part of the supplementary, and others are free to copy the gene names into Arabidopsis.org and validate our analysis.

However, in response to the reviewer's comment, we have removed the text introducing the analysis of a subset of 100 genes, and instead simply discuss the function and homology of interesting genes found to be under selection.

Minor issues:

#R1C11 - Figure 2a. Some chromosome IDs are different from the rest, such as 116, 1632, 1634, and 1633. You may consider renaming those or explaining the naming rules in the figure legends.

We agree. We have now changed the chromosome names.

#R1C12 - Line 273: "Fast evolutionary rates" sounds ambiguous. It is not very clear to know what the authors want to say here.

We have changed this to “Genome rearrangements in Heliantheae”.

#R1C13 - The authors mention “Supplementary information” several times throughout the manuscript, but I could not find the document in the review materials.

‘Supplementary information’ is just a reference to the supplementary figures or tables.

Reviewer #2 (Remarks to the Author):

#R2C1 - The authors assembled the genome of the Galapagos endemic, allotetraploid daisy species *Scalesia atractyloides* to identify the genomic basis of the island syndrome and the links between genome evolution and adaptive radiations on islands.

While the goals of the paper are interesting and the performed genome assembly and subsequent analyses generally solid, I think that the links between genomic characteristics and island radiations remain very speculative. A considerable part of the analyses is devoted to characterizing the two subgenomes of the allotetraploid species, and this is the stronger part of the analyses. However, the proposed links between genomic characteristics and the rapid, pronounced phenotypic changes associated with island life are tenuous and unsatisfactory. Additionally, I feel that the paper suffers from some degree of conceptual confusion about fundamental topics of island biology. For example, the paper remains vague about the concepts of island radiation vs. insular adaptive radiation, using the terms interchangeably, even though they are not. I will justify my assessment in greater detail below.

We are grateful with #R2 for their assessment of our work. Island biology has been characterised by a heavy focus on biogeography, macroevolution and macroecology. This genome represents one of the first attempts – if not the first – to search for a genomic basis of the island syndrome. We believe that the signatures found in terms of natural selection and gene expansion conform well to the expectations that have been described and predicted by systematists and natural historians over the last century. This said, we agree that comparative genomic analysis can be speculative, however, we see this paper as a hypothesis generator, and as a confirmation of overall patterns that can be subsequently functionally validated and confirmed, for instance, by functional studies enabled by e.g. CRISPR (to confirm the role of certain genes) and by a denser sampling (population-level sampling).

#R2C2 - L91: The concept of island syndrome must be better differentiated from the scenario of colonization of other ecological niches that are not on islands. In other words, novel selective pressure likely arise after colonization of any empty ecological space, whether it is on islands or not.

We agree, and have changed our working definition to:

“The ‘island syndrome hypothesis’ predicts the repeated and pronounced phenotypic shifts that some species undergo after colonising islands, as a result of a predictably different set of environmental conditions (Baeckens and Van Damme 2020).”

#R2C3 - L102-103: Explain how plants change as a result of island colonization, defining directionality of character change from mainland to island.

Thank you for allowing us to clarify. We have changed the text to clarify this. It is generally complicated to define directionality since what happens in the context of island evolution is that multiple phenotypes evolve, on multiple ways. For instance, Blonder et al 2016 explored leaf evolution in the silversword alliance and they concluded “that leaf functional trait variation (i) spans much of the global angiosperm range, (ii) is best explained by a white-noise evolutionary model and (iii) is integrated in ways consistent with both the global leaf economics spectrum and the predictions of leaf venation network theory.”.

In other words, it is hard to define directionality of character change, when so many [extreme] changes are in place.

#R2C4 - L112 (entire paragraph): The authors need to make a clear distinction between radiations and adaptive radiations. Not all radiations are adaptive. In adaptive radiations, diversification is driven by adaptation (for example to different habitats), rather than by allopatric speciation followed by adaptation and phenotypic divergence. This is an important conceptual distinction that does not come up in the paper. The authors seem to merge the concepts of radiations and adaptive radiations. A lot has been written on this distinction, starting from the classic book by Givnish and Sytsma on the topic of island radiations.

For example, in the studied group, the hypothesis of adaptive radiation would receive at least some degree of initial support if species occurring in different habitats on the same island evolved from a single common ancestor. Another way to put it is that the hypothesis of adaptive radiation would receive some tentative support if species co-occurring on the same island in different habitats were more closely related to each other than species occurring in the same (or similar) habitats on different islands. The authors should discuss these concepts and explain whether the adaptive nature of the studied radiation has been proven, for example, using the abovementioned phylogenetic approach. As the famous evolutionary biologist GC Williams noted already in 1966, the concept of adaptation is an onerous one, which is difficult to prove, so it is important that it is used very carefully and only if some analytical evidence is available.

The fact that this group of island plants presents variation of morphological features is a necessary but not sufficient condition to prove the adaptive nature of the radiation.

Additionally, the concept of radiation itself is often linked to the speed of diversification, so the authors should also explain whether previous evidence proves this point, typically through some sort of molecular dating analysis (or other means, if available).

If no analytical evidence is available to prove that the mentioned plant diversification is indeed an adaptive radiation, then the authors should refrain from using this term.

The reviewer is absolutely right, and we are grateful for their suggestion, which has made our abstract introduction much more fluid. We have removed references to *Scalesia* being a radiation, and focus on the island syndrome.

As we highlighted on #R1C2, we changed:

“Oceanic archipelagos comprise multiple disparate environments over small geographic areas and are isolated from other biotas. These conditions have led to some of the most **rapid and spectacular phenotypic changes, which are often repeated, thus offering a unique chance to characterise its genomic basis.** **These** repeated patterns of evolutionary change in plants on oceanic archipelagos, **or the** ‘plant island syndrome’, include changes in leaf **phenotypes**, acquisition of perennial life-style, flowering period and self-compatibility, and ancestral ploidy.”

In the introduction it reads:

“Organisms colonising these regions encounter highly distinct microenvironments that provide abundant ecological niches and thus ideal conditions for **rapid and pronounced phenotypic change** (Lomolino, Riddle, and Whittaker 2017).”

“Because the most prominent examples of **island syndromes feature animal lineages**, our understanding of these phenomena in plant lineages lags (Burns 2019). As plants colonise archipelagos, they typically and repeatedly undergo directional shifts in leaf morphology, dispersal ability, lifespan and size (Burns 2019). **This is well exemplified by the iconic, yet understudied, daisies in the genus *Scalesia*** (Blaschke and Sanders 2009; Fernández-Mazuecos et al. 2020; Crawford et al. 2009; U. Eliasson and U 1974). This group consists of ca. 15 species, which have colonised moist forests, littoral zones, arid zones, dry forests, volcanic soils, lava gravels and fissured environments across varied elevations (Itow 1995; Blaschke and Sanders 2009). The morphological changes undergone by *Scalesia* include an increased woodification, leaf-morphology variation, simplified inflorescences, increased growth rates, and gigantism - as expected by the plant island syndrome. Indeed, this outstanding phenotypic and ecological variation has led authors to refer to this group as the ‘Darwin finches of the plant world’ (Stöcklin 2009). All *Scalesia* species are ancestrally tetraploid ($2n=4x=68$) (Ono 1967; Uno Eliasson 1974), and the polyploid genetics may have provided the genetic grist for the diversification, as has been speculated for several island floras (Meudt et al. 2021).”

#R2C5 - L148-149: It seems like this is not an especially complete genome assembly. What is the percentage of the genome that was actually assembled?

The completeness of assemblies are typically evaluated with a set of conserved genes (BUSCO) - a set of highly curated genes expected to be conserved in all taxa. The *Scalesia atractyloides* has 97.1% of completeness, with only 17 out of 430 genes absent. With such a highly contiguous genome and high BUSCO completeness, these genes could indeed be biologically absent in this genome. However, we do agree that a substantial portion of the genome (~700 mb) has been condensed or lost in the final assembly. However, the focus on

BUSCO-genes suggests that the portion missing these are not gene-coding regions, and the genome's overall high repeat content further suggests that repeats were condensed by the assembly or purged from purge haplotypes. This suggests that the algorithms may have struggled to reconstruct a portion of the repeat regions.

#R2C6 - L208: Define what paleoallopolyploids are and the available evidence that the studied species is indeed one of them.

Thank you for this comment, we have now added to the text:

“The identification of differently represented transposable element families also provides compelling evidence that the *Scalesia* radiation is of allopolyploid origin, **confirming chromosome counts (Eliasson 1974)**. Island floras are characterised by a high degree of paleoallopolyploids (**i.e. old allopolyploids (Funk et al. 2009)**)”

#R2C7 - L315-316: Thus, the hypothesis of genome miniaturization potentially connected with island colonization in *Scalesia* is unwarranted, since the direction of change has not been established.

Even if this were the case, what would the adaptive value of genome miniaturization on islands be? If the authors wanted to establish a clear link between this genomic feature and adaptive island radiation, they would need to be much more specific about a potential mechanism.

We agree it was very speculative. We have removed this whole paragraph.

#R2C8 - L351-352: This evidence on leaf morphology is very tenuous.

Following the reviewer's concern, we now use a less specific term: 'leaf-associated'.

#R2C9 - L366-368: OK, but how is this linked to island life and island radiations?

We have added:

“This may underlie the natural history observations where *Scalesia* individuals growing without permanent light, **as provided by the open Galápagos landscapes**, show ”

#R2C10 - L371-372: But again, this would not be specific to dry conditions just on islands.

This is correct. But *Scalesia* lives in an arid environment, and arid environments are common in the context of islands.

#R2C11 - L381-382: I feel that the authors have not established a solid link between the characteristics of the target plant genome and island adaptive radiation. For example, they list a series of genes under selection with very generic functions, e.g. genes associated with growth and transitions between life stages. Are these genes under selection also in non-insular relatives of the studied species? It seems to me this kind of comparative genomic analyses of genes under selection in island plants versus non-insular relatives would be necessary to make a tentative case for the identification of links between genomic

characteristics and adaptations to island life and the proposed adaptive radiation of this groups of plants.

We agree with the reviewer:

1. We have now dropped references to adaptive radiation. These genes are explained in the light of the island syndrome, which we now contextualise better in the introduction.
2. Unfortunately the number of reference genome for South American plants lags and the closest related genome is that of the sunflower. We do agree that a closely related genome would improve our analyses, but we are limited by the data we have.

#R2C12 - L395-396: Very speculative and vague.

We thank the reviewer for this comment, and we have elected to remove this small paragraph relating to immunity and microbial defence.

#R2C13 - I found a few spelling or similar mistakes:

L78: specular instead of spectacular

L342: misspelled revolution for evolution

Figure 4 legend seems to have a lingering editing mistake.

All minor changes corrected.

Reviewer #3 (Remarks to the Author):

#R3C1 - This manuscript provides the most detailed assessment of genomic evolution in a plant lineage resulting from adaptive radiation in a remote archipelago. Several of the findings are noteworthy, especially considering the widespread incidence of ancestral, recent allopolyploidy in flowering plant lineages that have diversified extensively in oceanic islands. Beyond demonstrating a recent allopolyploid origin for *Scalesia*, which is unquestionably the most important example of adaptive radiation in the Galápagos flora, the findings of a lack of subgenome dominance in retention and expression of genes, considerable structural differentiation of the two subgenomes (by translocations and inversions) and high gene density within inverted regions, and characterization of many of the genes under positive selection as ones previously associated with functions that arguably may have been important in adaptive evolution are all major contributions of this study. In addition, the work is methodologically significant for using fossil transposable elements (TEs) to assign genes to subgenomes, which is an important challenge in polyploid plant genomics. The methodologies in general are sound, cutting-edge, and detailed sufficiently for replication. The results of this study will be of value to plant evolutionary biology in general, including phytogeography and plant evolutionary genomics – it is of broad interest. The analyses are sound and sufficient to back up the conclusions presented. The only minor concern worth mentioning here is that phylogenetic divergence time estimates should be indicated as approximate (using ca. or ~) unless error in the estimates are provided.

We are grateful for the extremely positive assessment of our work by #R3.

Below are suggested minor edits, indicated by line number on the submitted pdf of the manuscript

#R3C2 - Line 66: Change “change in ploidy” to “ancestral polyploidy”? As noted later in the manuscript, change in ploidy within remote archipelagos is rare and not truly part of the island syndrome – but polyploidy is associated with successful colonization and radiation, as a pre-condition rather than polyploidization after colonization (as in *Scalesia*)

78: spectacular (spelling)

Changed accordingly.

#R3C3 - 108: Need reference(s) for Juan Fernandez radiations

The reviewer is absolutely right and we apologise for overlooking. We added:

1. Takayama, K., López-Sepúlveda, P., Kohl, G., Novak, J. & Stuessy, T. F. Development of microsatellite markers in species of *Erigeron* (Asteraceae) endemic to the Juan Fernández Archipelago, Chile. *Am. J. Bot.* 99, 2010–2012 (2012).

2. Crawford, D. J., Stuessy, T. F., Cosner, M. B., Haines, D. W. & Silva, M. O. Ribosomal and chloroplast DNA restriction site mutations and the radiation of *Robinsonia* (Asteraceae: Senecioneae) on the Juan Fernandez Islands. *Plant Syst. Evol.* 184, 233–239 (1993).

3. Takayama, K. et al. Relationships and genetic consequences of contrasting modes of speciation among endemic species of *Robinsonia* (Asteraceae, Senecioneae) of the Juan Fernández Archipelago, Chile, based on AFLPs and SSRs. *New Phytol.* 205, 415–428 (2015).

#R3C4 - 115: Change “littoral” to “littoral zone” and “arid” to “arid zone” to avoid possible confusion that these are forest types rather than ecological zones

119: capitulum (spelling) AND change “The outstanding” to “This outstanding”

123: Change “for other island floras” to “for several island floras” (including Galápagos)

126: Change “most basal” to “one of the deepest” (that is, there are two deepest (sister) lineages in any clade, unless the base is unresolved)?

207: Change “of allopolyploid lineage” to either “of allopolyploid origin” or “an allopolyploid lineage”)

258: Add “in” before “paleopolyploid plant genomes”

All changed accordingly.

#R3C5 - 270 through 271: This same speculation about the importance of allopolyploidization in adaptation to insular environments was proposed by Barrier et al. (1999) Mol. Biol. Evol. upon discovery that the silversword alliance genome is allotetraploid

Thank you for the suggestion. We have now added Barrier et al 1999 as a reference.

#R3C6 - 301: Replace commas with decimal points in 693,2 and 586,5

Changed accordingly.

#R3C7 - 366 through 367: Does “absence of permanent light conditions” mean partially shaded conditions (meaning unclear)?

We have made this more clear now by adding:

“This may underlie the natural history observations where *Scalesia* individuals growing without permanent light conditions, as provided by the open Galápagos landscapes”

#R3C8 - 396: Change “encounter” to “encountered”

423: Change “flowering of plants” to “flowering plants”

424: Change “first and solid” to “first solid”

461: Change “chromatic” to “chromatin”

461 through 464: Change present-tense to past-tense for description of methods here

489: Change “detecting a removing ” to “detecting and removing”

490: Delete “and” after “input to”

492: Delete “was reduced”

561: Change “which” to “that”

570: Change “In specific” to “Specifically”

581 through 583: Best to delete “the” before common names of the four species of Asteraceae (i.e., change “the lettuce” to “lettuce”).

595 through 596: The sentence spanning these lines is incomplete

606: Change “which” to “that”

615: Change “insides” to “inside”

617: Change “which” to “that”

620: Change “not stringent” to “non-stringent”

628: A word is missing after “H. annuus” at end of sentence

650: Change “were” to “was”

661: Change “gene” to “genes”

670 through 671: Change to “Four of the six clusters ...” (that is, to avoid beginning sentence with a numeral)

All changed accordingly.

Reviewers' Comments:

Reviewer #1:

Remarks to the Author:

The authors have revised the manuscript. I really appreciate their response and revision, but I am afraid some of my previous questions, especially those on comparative genomics, have not been well solved or could use extra explanation.

The current Introduction is much clearer and better defines the scope of the study compared to the previous version. Some ambiguous terms have also been clearly explained, in my opinion. That being said, I would agree with Reviewer 2, who asked for some explanations about changes in characteristics between mainland and island species. Although the authors explained that it is difficult to define such characteristics in their response to Reviewer 2, I think it is pretty relevant to the island syndrome. If it is difficult to define such "syndromes", how could one study the genomic basis of such "syndromes"?

To distinguish the two sets of chromosomes, I understand that the authors first used COS to find homologous chromosomes, then split the two sets of chromosomes based on selected k-mer profiles of TEs. However, I am afraid I still have to disagree with the authors that showing SIX unevenly distributed TEs in the two subgenomes would support the classification or show the power of the assignment (Figure 2B), because it is very likely to find such TEs no matter how the two sets of chromosomes are assigned. Therefore, more sophisticated approaches incorporating the genome-scaled data should be applied here.

It is still an issue for dating the divergence of the two progenitors and the tetraploidization event. Using a single-point calibration is usually problematic, but the papers cited by the authors all used calibrations in the same way, so I think it is probably acceptable to the community. However, I would like to point out that the dating approach using TEs still lacks clarity here. Based on the papers cited by the authors, to date the tetraploidization event, one needs to find TEs that exist in both subgenomes with relatively high abundance, considering those TEs got inserted after the tetraploidization. By comparing the distance between the shared TEs and the distance between the shared TEs and their homologs in an outgroup, one can estimate the substitution rate based on the strict molecular clock and hence obtain an estimate of the date when the activity of shared TEs resumed, hence place a date for the tetraploidization. To date the divergence between the two progenitors, however, one needs to look for TEs that are specific to each subgenome and to estimate the ranges of their insertion times, i.e., how long the two progenitors were separated.

Based on the Methods, the authors used single-copy protein-coding genes to estimate the divergence between the two progenitors. Then they looked at the shared TEs and selected TEs that are monophyletic in *S. Atractyloides* and non-monophyletic for the two subgenomes to date the tetraploidization. However, I am totally lost with Supplementary Figure 4, as I could not understand where the numbers above the two distributions come from. I think Supplementary Figure 4 is probably an intermediate plot used to estimate the time of tetraploidization, but not a plot to justify the numbers in the main text and Figure 2C.

Lastly, in the previous version, the authors randomly selected 100 out of 920 positively selected genes. I can understand that it is labor-intensive to investigate all the 920 genes, but there might be other ways to select interesting genes. In the revision, the authors just removed the part saying that they randomly selected 100 genes. Nevertheless, those gene families are still randomly selected (Suppl. Table 07), and I am not all convinced that the randomly selected ones would be interesting. Also, referred to the comments from Reviewer 2 in the previous round, many of those genes are with generic functions. Without a non-insular relative of these species and clearly defined characteristics of island syndrome for daisies, the link between the plant island syndrome and the identified positive selected genes is still vague.

Minor comments:

Line 208-209: It would be better to tell the number of COS and the relevant reference used in identifying homologous chromosomes. Also, the sentence refers to Figure 2A, but the lines in the circos plot are not COS but aligned sequences from MUMMER, which are only mentioned in the Methods.

Figure 2C: there is no scale for the ultrametric tree.

Some figures are still using the old chromosome numbers, e.g., Suppl. Fig. 3.

Line 142: "the deepest lineage" is still not a correct term in phylogenetics. "A sister group to the rest of *Scalesia* species" sounds more correct.

Line 235: "which is line" -> "which is in line"?

Reviewer #4:

Remarks to the Author:

This is a very nice manuscript looking at the genome of the critically endangered *Scalesia* to understand the genomic basis of island syndrome in plants. There is a much needed demand in the plant genomics field to assemble the genomes of wild plants as many of our knowledge has been obtained by studying crop genomes. Naturally our understanding the basis of evolution from natural plants is far less understood, and I applaud the authors for generating a crucial and very high-quality resource for studying plant evolution and specifically addressing a long-standing question of island biology, the molecular basis of island syndrome. The genome assembly and subsequent annotation is very high quality and I have no major comment or question on that subject. Taking advantage of the high quality reference genome the authors have conducted several evolutionary genomic analysis to study the evolutionary origins of the Galapagos *Scalesia*. Through synteny and orthology analysis they find evidence of paleopolyploidy in *Scalesia* ancestral origin, and through selection analysis discover candidates genes that were selected and associated with the island syndrome. In the end I think this is a well analyzed manuscript using sophisticated methodologies to study the genome evolution and I have some comments that may or may not improve the manuscript that I wanted to suggest, but ultimately I'd recommend accepting with minor edits.

Comment1

I really like the TE annotation method the authors used for figuring out the subgenomes for *Scalesia*. I think having a high quality reference genome makes the TE distribution as a powerful method to determine the subgenome status and I can see this as a way many people may also use in the future as well. My only comment is in the dating of the genome duplication. People have commonly/traditionally looked at Ks values of paralogous genes to determine the number of duplication the timing of duplication etc, for instance these papers here:

<https://www.nature.com/articles/nature08670>

<https://academic.oup.com/plcell/article/16/7/1667/6010415>

And I think this is an alternative and conventional way people have been looking at duplication event and timing?

Comment 2

Its clear that there are two subgenomes for *Scalesia* meaning the two genomes have experienced different past evolutionary histories that are left over in each subgenome. But for the PSMC analysis

the authors are effectively combining the genomic variation from both subgenomes into one and conducting PSMC, and it seems a little bit odd to do this. Could they separate the variation by subgenome and redo the PSMC analysis to see the past demographic changes for each subgenome and maybe with that there could be new or better insight with their results.

Comment 3

This relates to comment 2 as well and its with the selection analysis. What's the evidence that this is positive selection and not relaxation of selection? With paleopolyploids it's well known there's subgenome dominance that happens and you see relaxation of selection in one of the subgenomes. I understand the method HYPHY they use is specifically testing for evidence of positive selection ($dN/dS > 1$) but are these signals coming from a specific subgenome? Or are there evidence that relaxation of selection happening in one subgenome is being detected as false positive adaptive evolution? Basically from their Fig3 there is some evidence of subgenome dominance that may be happening for *Scalesia* and I'm wondering how is this affecting their selection analysis?

Below we present a point-by-point response to the concerns raised by the two reviewers who commented on this version of the manuscript. In brief, we have:

1. Added Ks-plots as requested by reviewer #R4. This analysis confirmed our dating estimations.
2. Added detailed supplementary information on: *(i)* repeat numbers by subgenome and *(ii)* repeat families differentially represented in different subgenomes, as requested by #R1.
3. Added specifics regarding the *k*-mer analysis to the main text (M&M and results), in response to the concerns of #R1, as suggested by #R4.
4. Substantially rewrote, and improved the M&M and introduction by answering #R1's concerns.
5. Verified that the selection of 100 random genes did not bias the selection analyses and interpretation of findings.
6. Provided all the code in GitHub, which will ensure the reproducibility and accessibility of our work. See here: https://github.com/jcerca/Papers/tree/main/scalesia_genome
7. Uploaded the genome assembly at DRYAD (https://datadryad.org/stash/share/OUBW6bgTb3wDHNPdFMs_R06iQWLwfeaXHyVi1UE685c) and the raw reads at ENA (<https://www.ebi.ac.uk/ena/browser/view/PRJEB52418>). Because of the large amount of data (>1.1 Terabytes), these datasets will not become publicly available before a few days from 4 May, 2022.

We believe that addressing these issues has substantially improved our manuscript, and hope they meet the high standards of *Nature Communications*. We have added a note of appreciation to the reviewers in the Acknowledgements section.

José & Mike

REVIEWER COMMENTS

Reviewer #1 (Remarks to the Author):

The current Introduction is much clear and better defines the scope of the study compared to the previous version. Some ambiguous terms have also been clearly explained, in my opinion. That being said, I would agree with Reviewer 2, who asked for some explanations about changes in characteristics between mainland and island species. Although the authors explained that it is difficult to define such characteristics in their response to Reviewer 2, I think it is pretty relevant to the island syndrome. If it is difficult to define such “syndromes”, how could one study the genomic basis of such “syndromes”?

We thank the reviewer for following up with reviewing our paper. We appreciate that the reviewer noticed our changes to the Introduction, and thank them for their comments, which have further improved the Introduction. Regarding the expected changes in insular species, we have added further information. We have now clarified this, by adding the expectations regarding the island syndrome.

Additions are in bold, ~~removed text~~ is as strikethrough.

“ Because the most prominent examples of island syndromes feature animal lineages, our understanding of these phenomena in plants ~~lineages~~ lags (Burns 2019). As plants colonise archipelagos, they typically ~~and repeatedly~~ undergo shifts in leaf ~~phenotypes morphology~~, **overall size, woodiness, lifespan and have an altered dispersal ability**; ~~der lifespan and size - the plant island syndrome~~ (Burns 2019). ”

In the same paragraph:

“The ~~phenotypic morphological~~ changes undergone by *Scalesia* include increased **woodiness**, leaf-morphology variation, simplified inflorescences, increased growth rates, and gigantism - as expected for the plant island syndrome.”

To distinguish the two sets of chromosomes, I understand that the authors first used COS to find homologous chromosomes, then split the two sets of chromosomes based on selected k-mer profiles of TEs. However, I am afraid I still have to disagree with the authors that showing SIX unevenly distributed TEs in the two subgenomes would support the classification or show the power of the assignment (Figure 2B), because it is very likely to find such TEs no matter how the two sets of chromosomes are assigned. Therefore, more sophisticated approaches incorporating the genome-scaled data should be applied here.

We thank the reviewer for this comment, as it gives us yet another opportunity to improve the clarity of our writing and to further demonstrate the strength of our results. The six unevenly distributed TE families are simply examples selected to represent the overall pattern. However, given the reviewer’s concerns, we have now added a new supplementary table and a supplementary figure. Namely, we added *(i)* transposable element representation (family-level as determined by RepeatMasker) per subgenome in Supplementary Table 04, as well as *(ii)* details about subgenome-level differences in transposable element content. Specifically, for each repeat family, we calculated the ratio between transposable element representation in each subgenome. Supplementary Figure 05 is copied below. This shows that 52 families have a substantially different TE representation (defined as: one subgenome has 10 times more TEs of a given family than the other subgenome).

Supplementary Figure 05. Ratio of transposable element representation. For each family identified by RepeatMasker (*x*-axis), we plot the ratio (*y*-axis; subgenomeA divided by subgenomeB or subgenomeB divided by subgenomeA) between transposable elements for TE-number. 52 families had ratios bigger than 10, 17 families had ratios bigger than 100.

In addition to this, and to apply a more genome-scale approach, we randomized the chromosome-pairs (homeologs) using <https://www.random.org/lists/> and performed the subgenome separation analysis. We expected to obtain inconclusive results (i.e. no subgenome groupings) from this run because by randomizing the pairs we expect that no differentially represented TEs occur (because some pairs will be between chromosomes from the same subgenome, where no differentially represented TEs should occur). Indeed, by randomizing chromosomes we were unable to obtain results. This was added to the main text:

“Finally, to confirm whether *k*-mers separated both subgenomes reliably, we repeated the distance matrix and hierarchical clustering analyses with a slight modification: we randomized chromosome pairs. Under a random pairing we expected to obtain inconclusive results because chromosomes from the same subgenome should not have differentially represented TEs, and therefore subgenome groupings should not occur. Indeed, in line with our expectation, the randomization of the chromosome pairs yielded inconclusive results, as expected.”

It is still an issue for dating the divergence of the two progenitors and the tetraploidization event. Using a single-point calibration is usually problematic, but the papers cited by the authors all used calibrations in the same way, so I think it is probably acceptable to the community. However, I would like to point out that the dating approach using TEs still lacks clarity here. Based on the papers cited by the authors, to date the tetraploidization event, one needs to find TEs that exist in both subgenomes with relatively high abundance, considering those TEs got inserted after the tetraploidization. By comparing the distance between the shared TEs and the distance between the shared TEs and their homologs in an outgroup, one can estimate the substitution rate based on the strict molecular clock and hence obtain an estimate of the date when the activity of shared TEs resumed, hence place a date for the tetraploidization. To date the divergence between

the two progenitors, however, one needs to look for TEs that are specific to each subgenome and to estimate the ranges of their insertion times, i.e., how long the two progenitors were separated.

Based on the Methods, the authors used single-copy protein-coding genes to estimate the divergence between the two progenitors. Then they looked at the shared TEs and selected TEs that are monophyletic in *S. Atractyloides* and non-monophyletic for the two subgenomes to date the tetraploidization. However, I am totally lost with Supplementary Figure 4, as I could not understand where the numbers above the two distributions come from. I think Supplementary Figure 4 is probably an intermediate plot used to estimate the time of tetraploidization, but not a plot to justify the numbers in the main text and Figure 2C.

We thank the reviewer for identifying this weakness in the manuscript writing. After revising the text we came to agree that the text was not clear enough. We have now added clarifications on the M&M:

“We performed two separate dating analyses: one to date the nodes of the tree (speciation events) and another to date the polyploidization event. To date the nodes of the tree, we constrained ~~Dating of this tree was done by constraining~~ the node separating the *Scalesia* subgenomes and *Helianthus* at 6.14 Mya (Figure 2C) following recent literature (Mandel et al. 2019)(Fernández-Mazuecos et al. 2020)(Mandel et al. 2019). **Consistent with comparisons of Ks distributions (Supplementary Information; Supplementary Figure 05), a model-based divergence time estimate (r8s) suggests that the ancestral lineages represented in the *Scalesia* subgenomes diverged approximately 4.14 Mya. A second dating analysis was performed to date the polyploidization event using LTR retrotransposons. For this analysis, we used LTR retrotransposons which were evenly represented between subgenomes (to capture families active after the polyploidization event), and which were present in *Helianthus* (Supplementary Information; Supplementary Figure 06). By comparing genetic divergent (Jukes Cantor distances) between the *Scalesia* LTR retrotransposons and *Helianthus*, we estimated that the ancestral genomes reunited in a single polyploid genome at least 3.76 Mya** suggests that ~~This suggests that the subgenomes diverged from their MRCA roughly 4.14 Mya, but the separation of the ancestral lineages only lasted 0.5 My, as calculated by LTR-family divergence”~~

Some further clarifications.

- 1) While “one needs to find TEs that exist in both subgenomes” (copied from above), we do not necessarily need TEs “with relatively high abundance,”. What is strictly necessary is that TEs exist in equal numbers in both subgenomes. We have now clarified the M&M:

“Dating the polyploidization event was accomplished by combining the tree obtained by OrthoFinder (OrthoFinder run 1) and leaned on transposable element distributions along the subgenomes as previously detailed by ~~To date the divergence of the subgenomes we followed the approach of~~ (Session et al. 2016; Lovell et al. 2021; Mitros et al. 2020). Briefly, this approach has a simple assumption: before the speciation event (which separates the ancestral lineages) and after the polyploidization event (which brings the ancestral ~~lineages~~ **genomes** together), the accumulation of transposable elements will be similar on both subgenomes. **In other words, t** ~~r~~ **ransposable element families that are evenly** ~~equally~~ **represented on the both** subgenomes ~~will~~ therefore represent the pre-speciation and

post-allopolyploidization period. We focused on long-terminal repeats (LTRs) given their prevalence along the genome. **To obtain high-quality LTR sequences, we started by using** ~~We used~~ LTRharvest to identify LTR elements (Ellinghaus, Kurtz, and Willhoeft 2008), **followed by** ~~and~~ LTRdigest to process these elements. **LTRdigest annotated** ~~(i.e. annotating)~~ features such as genes inside LTRs, **and helped refine the elements**. To find ~~these~~ features **within the LTRs**, we downloaded various PFAM domains provided in (Steinbiss et al. 2009), **and complemented these by downloading and concatenating them with the** ~~downloaded~~ “Gypsy” and “Copia” domains from the PFAM online database. We converted the domains to HMMs using hmmconvert (Eddy 1992), and added HMMs from the Gypsy Database (Llorens et al. 2011). The identification and annotation of LTRs using these methods was done for the *S. atractyloides* and *Helianthus annuus* genomes, with the ~~the~~ **inclusion of the latter species serving as an outgroup for comparisons of genetic divergences. An important distinction relates to the LTR-element and the LTR-region: Instead of using the whole LTR-element involves the** ~~(i.e. whole transposable element including repeated regions and genes inside, while the)~~ **we used only the LTR-region involves only the Long Terminal Repeat of the LTR-element.** ~~We (long terminal repeat).~~ **For the next analyses we used only the LTR-region (as provided by LTR digest) as alignments were of better quality. Using LTR-domains of the *Scalesia* and sunflower genomes as inputs, we** ~~and ran~~ OrthoFinder **to obtain orthogroups consisting of** ~~group~~ closely related LTR-domains. We processed the OrthoFinder data **by selecting orthogroups that met two assumptions: 1) they were in equal representation on both subgenomes (as hypothesized above); 2) and that were also present in *Helianthus* (to calculate genetic distances, see below). Using orthogroups which met these two assumptions,** ~~After this,~~ we aligned the selected orthogroups using mafft, and cleaned poorly aligned regions using Gblocks (Castresana 2000; Talavera and Castresana 2007), with nont-stringent options **including** ~~(i.e. “allow smaller final blocks”, “allow gap positions within the final blocks”, and “allow less strict flanking regions”).~~ After this, **we further processed the data** by removing ~~ged~~ sequences with more than 50% missing data, and re-checked whether numbers of TEs were still balanced between subgenomes, **thus purging some further orthogroups.** We then re-aligned the data using mafft and inferred a tree for each ortholog. We kept only orthogroups where the *S. atractyloides* LTR-sequences were monophyletic, **but** ~~and~~ where both subgenomes were non-monophyletic. For the final set of 5 orthogroups passing all this filtering (Supplementary Figure 07) [NOTE: previously Supplementary Figure 04, that the reviewer was confused about], we calculated pairwise Jukes Cantor distances between each 1) *S. atractyloides* LTR-region, and between 2) *S. atractyloides* and *H. annuus*. The Jukes Cantor distances were plotted **as frequency histograms** in R (see Supplementary Figure 06), **and the peaks of the *Scalesia*-vs-*Scalesia* (golden on Supplementary Figure 06) and *Scalesia*-vs-*Helianthus* (grey on Supplementary Figure 06) were converted to** ~~we analysed the overall frequency and converted it to~~ million of years distance by a simple three rule with the *Helianthus* divergence with *Scalesia* of 6.14 Mya (Supplementary Information; Supplementary Figure 06).

- 2) “Based on the Methods, the authors used single-copy protein-coding genes to estimate the divergence between the two progenitors. Then they looked at the shared TEs and selected TEs that are monophyletic in *S. Atractyloides* and non-monophyletic for the two subgenomes to date the tetraploidization. However, I am totally lost with Supplementary Figure 4”

This is correct. To clarify this, we have changed the Supplementary Figure. (see answer above and the changed caption for Supplementary Figure 07, Previously Supplementary Figure 04.

“Supplementary Figure 07 Density distribution of pairwise comparisons of Jukes Cantor divergence We selected LTR-families evenly represented on the subgenomes, and present in the Helianthus genome for this analysis. These families were cleaned by removing missing data and cleaning the sequences using Gblocks. We ended up with 5 high-quality LTR-repeat families, which were used for dating the polyploidization event. In yellow, we plot *Scalesia* vs *Scalesia* LTRs, and in grey we plot *Scalesia* vs *Helianthus* LTRs. The arrows point to the the peak of the distribution, which was used to estimate the divergence between *Scalesia* subgenomes.”

Lastly, in the previous version, the authors randomly selected 100 out of 920 positively selected genes. I can understand that it is labor-intensive to investigate all the 920 genes, but there might be other ways to select interesting genes. In the revision, the authors just removed the part saying that they randomly selected 100 genes. Nevertheless, those gene families are still randomly selected (Suppl. Table 07), and I am not all convinced that the randomly selected ones would be interesting. Also, referred to the comments from Reviewer 2 in the previous round, many of those genes are with generic functions. Without a non-insular relative of these species and clearly defined characteristics of island syndrome for daisies, the link between the plant island syndrome and the identified positive selected genes is still vague.

We have explored genes under selection using three distinct, yet complementary approaches:

- First, we have done a GO enrichment analysis, which involves exploring and summarising all the genes.
- Second, we performed a STRING analysis, which involved exploring and summarising all the genes.
- Third, because we are interested in linking the phenotype of the island syndrome to the genotype, we have analysed the *Arabidopsis* literature for 100 genes. This is a novel way of integrating functional genomics from a model-organism with comparative genomics from a non-model organism. It is extremely laborious because it involves reading an average of 2-5 *Arabidopsis* papers per *Scalesia* ortholog. 100 genes thus represents an outstanding effort (> 100 working hours) from the lead author. This second tier is meant to complement the results from 1 and 2.

We agree with the reviewer that a selection of 100 genes (~15% of all the genes) under selection may bias results. To target the reviewer’s concern, we have done a GO enrichment analysis on the 100 randomly selected genes, and compared it to Figure 5A. By comparing 100 random genes with the total dataset and processing the data in a similar way, we are able to detect potential biases in sampling. As the images below show, we find no differences in the overall GO categories under selection (Figure 5A copied), and the Figure attached.

GO categories for the 920 genes under selection (Figure 5A).

A

GO categories for 100 random genes:

Some notes: The clusters are not organized in a similar way because REVIGO plots the clusters in arbitrary space. The clusters are very similar between both images and confirm that we have not biased the data.

We have now added to the manuscript:

“Before this analysis, we confirmed that the selection of 100 random genes did not bias the final results by comparing a GO analysis using the 920 genes (Figure 5A) with a GO analysis with only 100 genes. Subsampling did not bias the major categories.”

Finally, the comment: “ Without a non-insular relative of these species and clearly defined characteristics of island syndrome for daisies, the link between the plant island syndrome and the identified positive selected genes is still vague.”

We absolutely agree with the reviewer that the analysis of a closely related outgroup genome would increase the quality of our findings. However, high-quality genomes available across the tree of life, especially in areas such as South America, are currently limited. Because insular taxa encounter very contrasting environments, and likely undergo strong selective pressures to adapt, we think that our inclusion of the high-quality genome of a relatively close outgroup (*Helianthus*) yields trustworthy results from which we gleaned useful insights.

Minor comments:

Line 208-209: It would be better to tell the number of COS and the relevant reference used in identifying homologous chromosomes. Also, the sentence refers to Figure 2A, but the lines in the circos plot are not COS but aligned sequences from MUMMER, which are only mentioned in the Methods.

We have now changed the reference to Figure 2A.

Figure 2C: there is no scale for the ultrametric tree.

Changed accordingly.

Some figures are still using the old chromosome numbers, e.g., Suppl. Fig. 3.

Changed accordingly.

Line 142: “the deepest lineage” is still not a correct term in phylogenetics. “A sister group to the rest of Scalesia species” sounds more correct.

Changed accordingly.

Line 235: “which is line” -> “which is in line”?

Changed accordingly.

Reviewer #4 (Remarks to the Author):

This is a very nice manuscript looking at the genome of the critically endangered *Scalesia* to understand the genomic basis of island syndrome in plants. There is a much needed demand in the plant genomics field to assemble the genomes of wild plants as many of our knowledge has been obtained by studying crop genomes. Naturally our understanding the basis of evolution from natural plants in far less understood, and I applaud the authors for generating a crucial and very high-quality resource for studying plant evolution and specifically addressing a long-standing question of island biology, the molecular basis of island syndrome. The genome assembly and subsequent annotation is very high quality and I have no major comment or question on that subject. Taking advantage of the high quality reference genome the authors have conducted several evolutionary genomic analysis to study the evolutionary origins of the Galapagos *Scalesia*. Through synteny and orthology analysis they find evidence of paleopolyploidy in *Scalesia* ancestral origin, and through selection analysis discover candidates genes that were selected and associated with the island syndrome. In the end I think this is a well analyzed manuscript using sophisticated methodologies to study the genome evolution and I have some comments that may or may not improve the manuscript that I wanted to suggest, but ultimately I'd recommend accepting with minor edits.

We thank the reviewer for assessing our paper and for offering such kind and supportive words.

Comment 1

I really like the TE annotation method the authors used for figuring out the subgenomes for *Scalesia*. I think having a high quality reference genome makes the TE distribution as a powerful method to determine the subgenome status and I can see this as a way many people may also use in the future as well. My only comment is in the dating of the genome duplication. People have commonly/traditionally looked at Ks values of paralogous genes to determine the number of duplication the timing of duplication etc, for instance these papers here:

<https://www.nature.com/articles/nature08670>

<https://academic.oup.com/plcell/article/16/7/1667/6010415>

And I think this is an alternative and conventional way people have been looking at duplication event and timing?

Thank you for this comment. We agree that cross-validating our novel methods with more established methods will strengthen our manuscript and lend credibility to the analysis. We have now added Ks plots to the supplementary information (Supplementary Figure 06). The image is reproduced below after a brief explanation.

The Ks plots are consistent with Fig 2C in suggesting that the *Scalesia* duplicate blocks diverged shortly after the divergence of the *Scalesia* and *Helianthus* lineages. The left-most Ks peak for comparisons of the *Scalesia* subgenomes is 0.182 (top-most plot on the image copied below), quite close to the Ks peaks for the *Scalesia*-*Helianthus* comparisons (0.195-0.198). An estimate of the divergence between the *Scalesia* subgenomes based on simple calibration with the 6.14 MY *Scalesia*-*Helianthus* divergence time would be a bit older (~5.68 MY) than our previously estimated r8s-based estimate (4.14 MY). However, r8s corrects for among-lineage variation in substitution rates and thereby provides a more accurate estimation.

Change in the manuscript:

“ Consistent with comparisons of Ks distributions (Supplementary Information; Supplementary Figure 05), a model-based divergence time estimate (r8s) suggests that the ancestral lineages represented in the *Scalesia* subgenomes diverged approximately 4.14 Mya.”

Comment 2

It's clear that there are two subgenomes for *Scalesia* meaning the two genomes have experienced different past evolutionary histories that are left over in each subgenome. But for the PSMC analysis the authors are effectively combining the genomic variation from both subgenomes into one and conducting PSMC, and it seems a little bit odd to do this. Could they separate the variation by subgenome and redo the PSMC analysis to see the past demographic changes for each subgenome and maybe with that there could be new or better insight with their results.

This is an excellent question. Below we plot the PSMC for the whole *S. atrectyloides* genome, only subgenome A and only subgenome B. Both subgenomes have the same evolutionary history reflecting the relatively short time of divergent evolution before their reunification in the polyploid ancestor of *Scalesia*.

We have added this image as supplementary figure 03.

Supplementary Figure 03. PSMC for *Scalesia atrectyloides* and its subgenomes. The *y-axis* shows effective population size and the *x-axis* shows the years from present to the past.

Comment 3

This relates to comment 2 as well and its with the selection analysis. Whats the evidence that this is positive selection and not relaxation of selection? With paleopolyploids its well known there's subgenome dominance that happens and you see relaxation of selection in one of the subgenomes. I understand the method HYPHY they use is specifically testing for evidence of positive selection ($dN/dS > 1$) but are these signal coming from a specific subgenome?

For this analysis we are using ortholog sequences from high-quality (chromosome-resolved) Asteraceae genomes. We inferred positive selection using HYPHY's ABSrel model, using multiple sample correction. Because we used several sequences in the alignments, the model is able to distinguish between positive selection and relaxation of selection.

Or are there evidence that relaxation of selection happening in one subgenome is being detected as false positive adaptive evolution? Basically from their Fig3 there is some evidence of subgenome dominance that may be happening for *Scalesia* and I'm wondering how is this affecting their selection analysis?

This is an excellent question. We now report that in the 920 genes, 478 are on subgenome A and 442 on subgenome B. This shows no clear pattern of selection restricted to a specific subgenome.

We have now added this to the text:

“We identified 920 genes under selection ($p < 0.05$) in the *Scalesia* genome (**478 on subgenome A and 442 on subgenome B**)”

Reviewers' Comments:

Reviewer #1:

Remarks to the Author:

The authors have addressed each of the points I raised satisfactorily in their thoughtful revision.

Reviewer #4:

Remarks to the Author:

This is a substantially improved manuscript and I thank the authors for putting all that effort. I'm satisfied and I would really like to see it accepted soon.